# Inverse Preference Learning:
# Preference-based RL without a Reward Function

**Joey Hejna**
Stanford University
jhejna@cs.stanford.edu

**Dorsa Sadigh**
Stanford University
dorsa@cs.stanford.edu

## Abstract

Reward functions are difficult to design and often hard to align with human intent. Preference-based Reinforcement Learning (RL) algorithms address these problems by learning reward functions from human feedback. However, the majority of preference-based RL methods naïvely combine supervised reward models with off-the-shelf RL algorithms. Contemporary approaches have sought to improve performance and query complexity by using larger and more complex reward architectures such as transformers. Instead of using highly complex architectures, we develop a new and parameter-efficient algorithm, Inverse Preference Learning (IPL), specifically designed for learning from offline preference data. Our key insight is that for a fixed policy, the $Q$-function encodes all information about the reward function, effectively making them interchangeable. Using this insight, we completely eliminate the need for a learned reward function. Our resulting algorithm is simpler and more parameter-efficient. Across a suite of continuous control and robotics benchmarks, IPL attains competitive performance compared to more complex approaches that leverage transformer-based and non-Markovian reward functions while having fewer algorithmic hyperparameters and learned network parameters. Our code is publicly released[1].

## 1 Introduction

Reinforcement Learning (RL) has shown marked success in fixed and narrow domains such as simulated control [20] and game-playing [37]. When deploying RL in more complex settings, like in robotics or interaction with humans, one often runs into a critical bottleneck: the reward function. Obtaining reward labels in the real world can be complex, requiring difficult instrumentation [48, 57] and painstaking tuning [56] to achieve reasonable levels of sample efficiency. Moreover, despite extensive engineering, reward functions can still be exploited by algorithms in ways that do not align with human values and intents [21], which can be detrimental in safety-critical applications [5].

Instead of hand-designing reward functions, contemporary works have attempted to learn them through expert demonstrations [1], natural language [34], or human feedback [47, 3, 53]. Recently, reward functions learned through pairwise comparison queries—where a user is asked which of two demonstrated behaviors they prefer—have been shown to be effective in both control [13, 47, 32] and natural language domains [51]. This is often referred to as *Reinforcement Learning with Human Feedback (RLHF)*. Reward functions learned via RLHF can directly capture human intent, while avoiding alternative and more expensive forms of human feedback such as expert demonstrations. Preference-based RL algorithms for RLHF often interleave reward-learning from comparisons with off-the-shelf RL algorithms.

While preference-based RL methods discover reward functions that are aligned with human preferences, they are not without flaws. Learned reward functions must have adequate coverage of

---

[1] https://github.com/jhejna/inverse-preference-learning

both the state and action space to attain good downstream performance. Consequently, learning the reward function can be expensive, usually requiring thousands of labeled preference queries. To mitigate these challenges, recent works have proposed improving learned reward functions by adding inductive biases before optimization with RL. Hejna and Sadigh [22] pretrain reward functions with meta-learning. Park et al. [43] use data augmentation. Early et al. [15] and Kim et al. [28] make the reward function non-Markovian using recurrent or large transformer sequence model architectures respectively. Such approaches increase the upfront cost of preference-based RL by using additional data or compute. Moreover, these techniques still combine reward optimization with vanilla RL algorithms. Ultimately, this just adds an extra learned component to already notoriously delicate RL algorithms, further increasing hyper-parameter tuning overhead. Preference-based RL approaches often end up training up to four distinct neural networks independently: a critic (with up to two networks), an actor, and a reward function. This can be problematic as prediction errors cascade from the reward function, to the critic, and ultimately the actor causing high variance in downstream performance. To address these issues, we propose a parameter-efficient algorithm specifically designed for preference-based RL that completely eliminates the need to explicitly learn a reward function. In doing so, we reduce both complexity and compute cost.

The key insight of our work is that, under a fixed policy, the $Q$-function learned by off-policy RL algorithms captures the same information as the learned reward function. For example, both the $Q$-function and reward function encode information about how desirable a state-action pair is. This begs the question: why do we need to learn a reward function in the first place? Our proposed solution, Inverse Preference Learning or IPL, is an offline RL algorithm that is specifically designed for learning from preference data. Instead of relying on an explicit reward function, IPL directly optimizes the implicit rewards induced by the learned $Q$-function to be consistent with expert preferences. At the same time, IPL regularizes these implicit rewards to ensure high-quality behavior. As a result, IPL removes the need for a learned reward function and its associated computational and tuning expense.

Experimentally, we find that even though IPL does not explicitly learn a reward function, it achieves competitive performance with complicated Transformer-based reward learning techniques on offline Preference-based RL benchmarks with real-human feedback. At the same time, IPL consistently exhibits lower variance across runs as it does not suffer from the errors associated with querying a learned reward model. Finally, under a minimal parameter budget, IPL is able to outperform standard preference-based RL approaches that learn an explicit reward model.

## 2   Related Work

Our work builds upon literature in reward learning, preference-based RL, and imitation learning.

**Reward Learning.** Due to the challenges associated with designing and shaping effective reward signals, several works have investigated various approaches for learning reward functions. A large body of work uses inverse RL to learn a reward function from expert demonstrations [1, 41, 46], which are unfortunately difficult to collect [27, 2, 35] or often misaligned with true human preferences [7, 31]. Subsequently, reward learning techniques using other simpler forms of feedback such as scalar scores [29] and partial [38] or complete rankings [11, 8] have been developed. One of the simplest forms of human feedback is pairwise comparisons, where the user chooses between two options. Often, pairwise comparison queries are sampled using techniques from active learning [47, 9, 14]. However, to evaluate learned reward functions, these methods rely on either RL or traditional planning algorithms which are complex and computationally expensive. Our approach takes a simpler perspective that is parameter-efficient by combining reward and policy learning. Though it is not the focus of our work, IPL could additionally leverage active learning techniques for selecting preference data online.

**Preference-based Deep Reinforcement Learning.** Current approaches to preference based deep RL train a reward function, and then use that reward function in conjunction with a standard reinforcement learning algorithm [13, 33, 50]. Several techniques have been developed to improve the learned reward function, such as pre-training [25, 32], meta-learning [22], data augmentation [43], and non-Markovian modeling. Within the family of non-Markovian reward modeling [6], recent approaches have leveraged both LSTM networks [15] and transformers [28] for reward learning. But, these methods still rely on Markovian offline RL algorithms such as Implicit Q-Learning (IQL) [30] for optimization. Ultimately, this makes such approaches theoretically inconsistent as the policy learning

component assumes the reward to be only a function of the current state and action. All techniques for learning the reward function in combination with standard RL methods [20, 49] end up adding additional hyper-parameter tuning and compute cost. IPL on the other hand, is directly designed for RL from preference data and eliminates the reward network entirely. Other recent works also consider contrastive objectives instead of RL [26, 23].

Recently, works in natural language processing have applied ideas from preference-based RL to tasks such as summarization [51, 54], instruction following [42], and question-answering [40]. The RLHF paradigm has proven to be powerful even at the massive scale of aligning large language models. In this regime, learned reward models are massive, making an implicit reward method like IPL more attractive. In fact, IPL in a contextual bandits setting recovers concurrent work by Rafailov et al. [45] on implicit reward modeling in LLMs (see Appendix A). While we focus on control in our experiments, we hope our work can inform future explorations in language domains.

**Imitation Learning**. Our work builds on foundational knowledge in maximum entropy (MaxEnt) RL [58] and inverse RL [59]. Recent works in MaxEnt inverse RL have used the mapping between $Q$-functions and reward functions under a fixed policy. Specifically, Garg et al. [18] show that the regularized MaxEnt inverse RL objective from Ho and Ermon [24] can be re-written using the $Q$-function instead of a reward function and Al-Hafez et al. [4] stabilize their approach. While the relationship between $Q$-functions and rewards has been used for MaxEnt inverse RL, we study this relationship when learning from preference data. While both problems seek to learn models of expert reward, the data differs significantly — preference-based RL uses comparisons instead of optimal demonstrations. This necessitates a greatly different approach.

## 3 Inverse Preference Learning

In this section, we first describe the preference-based RL problem. Then, we describe how, leveraging techniques from imitation learning, we can remove the independently learned reward network from prior methods. This results in a simpler algorithm with lower computational cost and variance in performance.

### 3.1 Preference-Based RL

We consider the reinfrocement leraning (RL) paradigm where an agent seeks to maximize its expected cumulative discounted sum of rewards in a Markov Decision Process (MDP). Standard off-policy RL algorithms, do so using state, action, reward, and next state tuples $(s, a, r, s')$. In preference-based RL, however, the reward function $r$ is unknown, and must be learned from human feedback. Thus, Traditional preference-based RL methods are thus usually separated into two stages: first, reward learning, where $r_E$ is estimated by a learned reward function $r_\theta$, and second, reinforcement learning, where a policy $\pi(a|s)$ is learned to maximize $\mathbb{E}_\pi [\sum_{t=0}^\infty \gamma^t r_\theta(s, a)]$ with $\gamma$ as the discount factor. Though our method combines these two phases, we use the building blocks of each and consequently review them here.

**Preference Learning.** First, similar to prior works [13, 32], we assume access to preference data in the form of binary comparisons. Each comparison is comprised of two behavior segments, $\sigma^{(1)}$ and $\sigma^{(2)}$, and a binary label $y$ indicating which of the two was preferred by an expert. As in Wilson et al. [53], each behavior segment is simply a snippet of a trajectory of length $k$, or $\sigma = (s_t, a_t, s_{t+1}, a_{t+1}, \ldots, a_{t+k-1}, s_{t+k})$. Increasing $k$ can provide more information per label at the cost of potentially noisier labels. The label $y$ for each comparison is assumed to be generated by an expert according to a Bradley-Terry Preference model [10]:

$$P_{r_E}[\sigma^{(1)} \succ \sigma^{(2)}] = \frac{\exp \sum_t r_E(s_t^{(1)}, a_t^{(1)})}{\exp \sum_t r_E(s_t^{(1)}, a_t^{(1)}) + \exp \sum_t r_E(s_t^{(2)}, a_t^{(2)})}, \tag{1}$$

where $r_E(s_t, a_t)$ is again the expert's unknown underlying reward model. We use the subscript $r_E$ on probability $P$ to indicate that the preference distribution above results from the expert's reward function. Let the dataset of these preferences be $\mathcal{D}_p = \{(\sigma^{(1)}, \sigma^{(2)}, y)\}$. To learn $r_E$, prior works in preference-based RL estimate a parametric reward function $r_\theta$ by minimizing the binary-cross-entropy over $\mathcal{D}_p$:

$$\mathcal{L}_P(\theta) = -\mathbb{E}_{\sigma^{(1)}, \sigma^{(2)}, y \sim \mathcal{D}_p} \left[ y \log P_{r_\theta} \left[ \sigma^{(1)} \succ \sigma^{(2)} \right] + (1-y) \log \left( 1 - P_{r_\theta} \left[ \sigma^{(1)} \succ \sigma^{(2)} \right] \right) \right]. \tag{2}$$

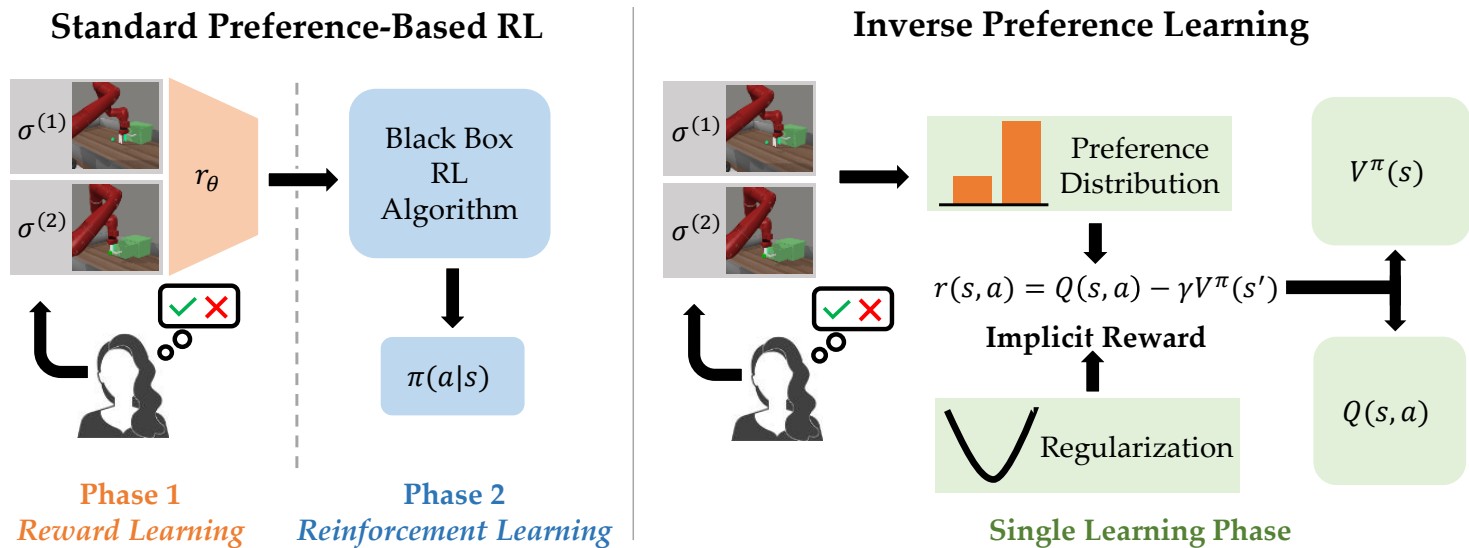

Figure 1: A depiction of the difference between standard preference-based RL methods and Inverse Preference Learning. Standard preference-based RL first learns a reward function, then optimizes it with a blockbox RL algorithm. IPL trains a $Q$ function to directly fit the expert's preferences. This is done by aligning the implied reward model with the expert's preference distribution and applying regularization.

This objective results from simply minimizing $\mathbb{E}_{\mathcal{D}_p}[D_{KL}(P_{r_E}||P_\theta)]$, the KL-divergence between the expert preference model and the one induced by $r_\theta$, effectively aligning it with the expert's preferences. We note that some other works in preference-based RL focus on learning an improved model $r_\theta$ to address the reward learning part of the problem [43, 28]. However, these methods still use off-the-shelf RL algorithms for the policy learning part of the problem.

**Reinforcement Learning.** Common off-policy RL methods learn a policy $\pi$ by alternating between policy evaluation (using the contractive Bellam Operator $\mathcal{B}_r^\pi$) to estimate $Q^\pi$ and policy improvement, where the policy $\pi$ is improved [52]. Concretely, after repeated application of $\mathcal{B}_r^\pi$ as

$$(\mathcal{B}_r^\pi Q)(s,a) = r(s,a) + \gamma \mathbb{E}_{s' \sim p(\cdot|s,a)}[V^\pi(s')], \qquad (3)$$

the policy can be improved by maximizing $Q$. In some settings, the Bellman operator $\mathcal{B}_r^*$ corresponding to the optimal policy $\pi^*$ can be used directly, removing the need for the policy improvement step. In these cases, we can simply extract $\pi^*$ from the resulting $Q^*$.

To learn the optimal policy, two-phase preference based RL methods rely on recovering the optimal $r_E$ in the reward learning phase before running RL. This potentially propagates errors from the estimated $r_\theta$ to the learned $Q$-function and ultimately learned policy $\pi$. In practice, it would be more efficient to eliminate the need for two separate stages. In the next section, we show how this can be done by establishing a bijection between reward functions $r$ and $Q$-functions.

### 3.2 Removing The Reward Function

In this section, we formally describe how the reward function can be removed from offline preference-based RL algorithms. Our key insight is that the $Q$-function learned by off-policy RL algorithms in fact encodes the same information as the reward function $r(s,a)$. Consequently, it is unnecessary to learn both. First, we show how the reward function can be re-written in terms of the $Q$ function allowing us to compute the preference model $P_Q$ induced by the $Q$-function. Then, we derive an objective that simultaneously pushes $Q$ to fit the expert's preferences while also remaining optimal.

Consider fitting a $Q$ function via the Bellman operator $\mathcal{B}_r^\pi$ for a fixed policy $\pi$ until convergence where $\mathcal{B}_r^\pi Q = Q$. Here, to encode the cumulative discounted rewards when acting according to the policy, the $Q$-function depends on both $r$ and $\pi$. This dependence, however, is directly disentangled by the Bellman equation. By rearranging it (Eq. (3)), we can solve for the reward function in terms of $Q$ and $\pi$. This yields the so-called inverse soft-Bellman operator:

$$(\mathcal{T}^\pi Q)(s,a) = Q(s,a) - \gamma \mathbb{E}_{s'}[V^\pi(s')]. \qquad (4)$$

In fact, for a fixed policy $\pi$ the inverse-Bellman operator is bijective, implying a one-to-one correspondence between the $Q$ function and the reward function. Though this was previously shown in maximum entropy RL [18], we prove the general case Lemma 1 in Appendix A.

Intuitively, this makes sense: when holding the policy constant, only the reward function affects $Q$. We abbreviate the evaluation of $(\mathcal{T}^\pi Q)(s, a)$ as $r_{Q^\pi}(s, a)$ to indicate that $r_{Q^\pi}$ is the unique implicit reward function induced by $Q^\pi$. Prior works in imitation learning leverage the inverse soft-Bellman operator to measure how closely the implicit reward model $r_{Q^\pi}$ aligns with expert demonstrations [18]. Our key insight is that this equivalence can also be used to directly measure how closely our $Q$ function aligns with the expert preference model *without ever directly learning $r$*.

Consider the Bradley-Terry preference model in Equation (1). For a fixed policy $\pi$ and its corresponding $Q^\pi$, we can obtain the preference model of the implicit reward function $P_{Q^\pi}[\sigma^{(1)} > \sigma^{(2)}]$ by substituting the inverse Bellman operator into Equation (1) as follows:

$$P_{Q^\pi}[\sigma^{(1)} > \sigma^{(2)}] = \frac{\exp \sum_t (\mathcal{T}^\pi Q)(s_t^{(1)}, a_t^{(1)})}{\exp \sum_t (\mathcal{T}^\pi Q)(s_t^{(1)}, a_t^{(1)}) + \exp \sum_t (\mathcal{T}^\pi Q)(s_t^{(2)}, a_t^{(2)})}. \tag{5}$$

This substitution will allow us to measure the difference between the preferences implied by $Q^\pi$ and those of the expert. To minimize the difference, we can propagate gradients through the preference modeling loss (Equation (2)) and the implicit preference model $P_{Q^\pi}$ (Equation (5)) to $Q$—just as we would for a parameterized reward estimate $r_\theta$. Unfortunately, naïvely performing this substitution is insufficient to solve the RL objective for two reasons.

**The Optimal Inverse Bellman Operator.** First, we have used an arbitrary policy $\pi$, not the optimal one, for converting from $Q$-values to rewards. Though the $Q$-function may imply the expert's preferences, the corresponding policy could be extremely sub-optimal. To fix this problem, we need to use the optimal inverse bellman operator $\mathcal{T}^*$ to ensure the extract $Q$-function corresponds to that of $\pi^*$. For this step, we can use any off-policy RL-algorithm that converges to the optimal policy! If the algorithm directly estimates the $\mathcal{B}_r^*$, the corresponding $\mathcal{T}^*$ can be estimated using the target from $\mathcal{B}_r^*$, or

$$(\mathcal{T}^* Q)(s, a) = Q(s, a) - \gamma \mathbb{E}_{s'}[V^{\text{targ}}(s')] \text{ where } V^{\text{targ}}(s) \text{ is estimated as in } \mathcal{B}_r^*.$$

In many cases, however, computing the optimal bellman operator $\mathcal{B}_r^*$ is infeasible. Instead, many modern off-policy RL algorithms use policy improvement to converge to the optimal policy. These methods, like Haarnoja et al. [20] use $Q^\pi$ to estimate a new policy $\pi'$ such that $Q^{\pi'} \geq Q^\pi$. By repeatedly improving the policy, they eventually converge to $Q^*$. Thus, by repeatedly improving the policy according to these algorithms, we can eventually converge to the optimal policy and can thus estimate corresponding optimal inverse bellman operator by using $V^{\text{targ}}(s) = \mathbb{E}_{a \sim \pi(\cdot|s)}[Q(s, a)]$ in the above equation.

**Regularization.** Given we can estimate $\mathcal{T}^*$ using targets from $\mathcal{B}_r^*$ or policy improvement, we can fit the optimal $Q$-function by minimizing the following loss function

$$\mathcal{L}_p(Q) = -\mathbb{E}_{\sigma^{(1)}, \sigma^{(2)}, y \sim \mathcal{D}_p} \left[ y \log P_{Q^*}[\sigma^{(1)} > \sigma^{(2)}] + (1 - y) \log(1 - P_{Q^*}[\sigma^{(1)} > \sigma^{(2)}]) \right].$$

where $P_{Q^*}$ is given by substituting $\mathcal{T}^* Q$ into Eq. (5). Unfortunately, optimizing this objective alone leads to poor results and may not converge when using RL algorithms that depend on policy improvement. This is because the above objective is under constrained due to the invariance of the Bradley-Terry preference model to shifts. By examining Eq. (1), it can be seen that adding a constant value to all rewards does not change the probability of preferring a segment. However, shifting the reward function by a constant *does* change the $Q$-function. RL algorithms using policy improvement monotonically increase the $Q$-function until reaching the maximum at $Q^*$. Thus as the implicit reward continues to increase, $Q^*$ will continue to increase and may never be reached. To resolve this issue, we insure that the optima of the preference loss is unique by introducing a convex regularizer $\psi(\cdot)$ on the implicit rewards $r_{Q^\pi} = \mathcal{T}^\pi Q$, giving us the regularized preference loss:

$$\mathcal{L}_p(Q) = -\mathbb{E}_{\sigma^{(1)}, \sigma^{(2)}, y \sim \mathcal{D}_p} \left[ y \log P_{Q^*}[\sigma^{(1)} > \sigma^{(2)}] + (1 - y) \log(1 - P_{Q^*}[\sigma^{(1)} > \sigma^{(2)}]) \right] + \lambda \psi(\mathcal{T}^* Q) \tag{6}$$

In practice we choose $\psi$ to be a form of L2 regularization as is commonly done in imitation learning [18, 4] to prevent unbounded reward values. $\lambda > 0$ is a hyperparameter that controls the strength of regularization. Besides allowing us to guarantee convergence, regularization has a number of benefits. It can help center the implicit reward near zero, which has been shown to beneficial for RL [16]. Moreover, it encourages more realistic implicit rewards. For example, a reward function might change rapidly by large values when only small perturbations are applied to the state or action. Though such reward functions might be unrealistic, they are completely valid solutions of the inverse-Bellman

operator. Adding regularization can help penalize large deviations in reward unless they drastically reduce the preference loss. Thus, the first term of Eq. (6) encourages the $Q$-function to match the expert's preferences, while the second term smooths the implied reward function and makes it unique.

Our final algorithm, which we call Inverse Preference Learning (IPL) fits the optimal policy corresponding to the regularized expert reward function by repeatedly minimizing $\mathcal{L}_p(Q)$ (Eq. (6)) and improving the value target used $V^{\text{targ}}$ with the update step from any off-policy RL algorithm. In this manner, IPL performs dynamic programming through the inverse bellman operator until convergence. In Appendix A, we prove the following Theorem.

**Theorem 1** *Given an off-policy RL algorithm that convergences to the optimal policy $\pi_r^*$ for some reward function $r$ and regularizer $\psi$ such that Eq. (2) is strictly convex, IPL converges to $\pi_{r^*}^*$ corresponding to reward function $r^* = \arg\min_r \mathbb{E}_{\mathcal{D}_p}[D_{KL}(P_{r_E}||P_\theta)] + \lambda\psi(r)$.*

The proof of the theorem essentially relies on the fact that for a fixed policy $\pi$, we can optimize $\mathcal{L}_p(Q)$ (Eq. (6)) to fit $r^*$. Then, we can update the policy (or target values $V^{\text{targ}}$) and optimize $\mathcal{L}_p(Q)$ again. Because $r^*$ is unique, we fit $r^*$ again the second time, but the $Q$-function has improved. There are many choices of regularizers where this holds. In tabular settings if $\psi(r) = r^2$, $\mathcal{L}_p(Q)$ reduces to L2 regularized logistic regression, which is strictly convex, guaranteeing convergence (Appendix A).

Effectively, IPL removes the need to learn a reward network, while still converging to similar solution as other preference-based RL algorithms. Learning a reward network requires more parameters and a completely separate optimization loop, increasing compute requirements. Moreover, an explicit reward model introduces a whole new suite of hyper-parameters that need to be tuned including the model architecture, capacity, learning rate, batch size, and stopping criterion. In fact, because human preference data is so difficult to collect, many approaches opt to use simple accuracy thresholds instead of validation criteria to decide when to stop training $r_\theta$ [32]. All of these components make preference-based RL unreliable and high-variance. On the other hand, our method completely removes all of these parameters in exchange for a single $\lambda$ hyper-parameter that controls the regularization strength. Though we have theoretically derived IPL, in the next section we provide practical recipes for applying it to offline preference-based RL.

### 3.3 IPL for Offline Preference-based RL

In offline preference-based RL, we assume access to a fixed offline dataset $\mathcal{D}_o = \{(s, a, s')\}$ of interactions without reward labels generated by a reference policy $\mu(a|s)$ in addition to the preference dataset $\mathcal{D}_p$. Common approaches to offline RL seek to learn *conservative* policies that do not stray too far away from the distribution of data generated by $\mu(a|s)$. This is critical to prevent the policy $\pi$ from reaching out-of-distribution states during deployment which can be detrimental to performance. In this section, we detail a practical version of IPL that uses the

---

**Algorithm 1:** IPL Algorithm (XQL Variant)

**Input :** $\mathcal{D}_p, \mathcal{D}_o, \lambda, \alpha$
**for** $i = 1, 2, 3, ...$ **do**
    Sample batches $B_p \sim \mathcal{D}_p, B_o \sim \mathcal{D}_o$
    Update $Q$: $\min_Q \mathbb{E}_{B_p}[\mathcal{L}_p(Q)]$ (Eq. (6))
    Update $V$: $\min_V \mathbb{E}_{B_p \cup B_o}[e^z - z - 1]$
        where $z = Q(s, a) - V(s))/\alpha$
Finally, extract $\pi(a|s)$ via:
    $\max_\pi \mathbb{E}_{\mathcal{D}_p \cup \mathcal{D}_o}[e^{(Q(s,a) - V(s))/\alpha} \log \pi(a|s)]$

---

$\mathcal{X}$QL offline RL algorithm [19]. $\mathcal{X}$QL fits the KL-constrained RL objective

$$\max_\pi \mathbb{E}_\pi \left[ \sum_{t=t'}^{\infty} \gamma^t \left( r(s_t, a_t) - \alpha \log \frac{\pi(a_t|s_t)}{\mu(a_t|s_t)} \right) \right]$$

where $\alpha$ controls the magnitude of the KL-divergence penalty. The $\mathcal{X}$QL algorithm directly fits the optimal $Q$-function using the optimal soft-Bellman operator [19, 55]

$$(\mathcal{B}_r^* Q)(s, a) = r(s, a) + \gamma \mathbb{E}_{s'}[V^{\text{targ}}(s')], \text{ where } V^{\text{targ}}(s) = \alpha \log \mathbb{E}_{a \sim \mu(\cdot|s)} \left[ e^{Q(s,a)/\alpha} \right].$$

In practice, $V^{\text{targ}}$ is estimated using the linex loss function over the current $Q$-function. Thus, to fit the optimal $Q$-function, IPL with $\mathcal{X}$QL alternates between minimizing the preference loss $\mathcal{L}_p(Q)$ (Eq. (6)), and updating a learned value function $V$ until they converge to $Q^*$ and $V^*$. Note that we are not limited to using just $\mathcal{D}_p$. Though the preference modeling part of $\mathcal{L}_p(Q)$ can only be optimized with preference data $\mathcal{D}_p$, the value function can be updated with offline data as well. In the presence

of additional offline data, we find that updating the value function using $\mathcal{D}_p \cup \mathcal{D}_o$ leads to better performance. We approximate L2 regularization with the regularizer $\psi(r) = \mathbb{E}_{\mathcal{D}_p \cup \mathcal{D}_o}[r(s,a)^2]$, which imposes an L2 penalty across the support of the data. While one might try to use weight decay to emulate L2-regularization, doing so is difficult in practice as $\mathcal{T}^*Q$ depends on both the $Q$ network and the target network. We find that weighting the regularization equally between $\mathcal{D}_p$ and $\mathcal{D}_o$ performs well. After $Q$ and $V$ have converged, we can extract the policy using the closed form relationship $\pi^*(a|s) \propto \mu(a|s) \exp((Q^*(s,a) - V^*(s))/\alpha)$ for KL-constrained RL as in Garg et al. [19], Peng et al. [44]. The full algorithm for IPL with $\mathcal{X}$QL can be found in 1.

Though we have shown how IPL can be instantiated with $\mathcal{X}$QL, it is fully with other offline RL algorithms. In fact, IPL can also be used with online RL algorithms like SAC [20]. Critically, this makes the IPL framework general, as it can remove the need for reward modeling in nearly any preference-based RL setting. This makes IPL simpler and more efficient. In the next section, we show that IPL can attain the same performance as strong offline preference-based RL baselines, without learning a reward network.

## 4 Experiments

In this section, we aim to answer the following questions: First, how does IPL compare to prior preference-based RL algorithms on standard benchmarks? Second, how does IPL perform in extremely data-limited settings? And finally, how efficient is IPL in comparison to two-phase preference-based RL methods?

### 4.1 Setup

As discussed in the previous section, though we use a KL-constrained objective for our theoretical derivation, in practice we can construct versions of IPL based on any offline RL algorithm. In our experiments we evaluate IPL with Implicit Q-Learning (IQL) [30], since it has been used in prior offline preference-based RL works. This allows us to directly compare IPL by isolating its implicit reward component and using the same exact hyper-parameters as prior works. Using IPL with IQL amounts to updating the value function according to the asymmetric expectile loss function instead of the linex loss function. Concretely, this can be done by replacing the value update in Algorithm 1 with $\min_V \mathbb{E}_{B_p \cup B_o}\left[|\tau - \mathbb{1}(Q(s,a) - V(s) < 0)|(Q(s,a) - V(s))^2\right]$ where $\tau$ is the expectile.

Inspired by Park et al. [43], we introduce data augmentations that sample sub-sections of behavior segments $\sigma$ during training. While such augmentations are inapplicable to non-Markovian reward models, we find that they boost performance for Markovian reward models while also reducing the total number of state-action pairs per batch of preference data. This is important as IPL needs data from both $\mathcal{D}_p$ and $\mathcal{D}_o$ to regularize the implicit reward function. Additional experiment details and hyper-parameters can be found in the Appendix.

### 4.2 How does IPL perform on preference-based RL benchmarks?

We compare IPL to other offline preference-based RL approaches on D4RL Gym Locomotion [17] and Robosuite robotics [36] datasets with real-human preference data from Kim et al. [28]. We compare IQL-based IPL, with the same hyper-parameters, to various baselines that learn a reward model $r_\theta$ before optimization with IQL. Markovian Reward or MR denotes using a standard Markovian MLP reward model, like those used in Christiano et al. [13] and Lee et al. [32]. Note that this is also equivalent to T-REX [11] for offline RL. Non-Markovian Reward or NMR denotes using the non-Markovian LSTM based reward model from Early et al. [15]. Preference Transformer (PT) is a state-of-the-art approach that leverages a large transformer architecture to learn a non-Markovian reward and preference weighting function. B-REX uses bayesian optimization to fit a linear reward function from predefined features [12], which in our case are random Gaussian projections of the states and actions. For fairness, we also compare against our own implementation of IQL with a Markovian Reward function that uses the same data augmentation as IPL.

Our results are summarized in Table 1. Starting with the first column, we see that preference-based RL methods are able to match IQL with the ground truth reward function in many cases. On, several tasks, however, the MR implementation from Kim et al. [28] fairs rather poorly. The non-Markovian methods, (NMR and PT) improve performance. It is worth noting that on many tasks our

| Dataset | IQL (Oracle) | MR (from [28]) | LSTM (from [28]) | PT (from [28]) | BREX (reimpl.) | MR (reimpl.) | IPL (Ours) |
|---|---|---|---|---|---|---|---|
| hop-m-r | 83.06 ±15.8 | 11.56 ±30.3 | 57.88 ±40.6 | **84.54 ±4.1** | 62.0 ±20.3 | 70.20 ±35.0 | 73.57 ±6.7 |
| hop-m-e | 73.55 ±41.5 | 57.75 ±23.7 | 38.63 ±35.6 | 68.96 ±33.9 | 85.1 ±8.0 | **103.0 ±5.6** | 74.52 ±10.1 |
| walk-m-r | 73.11 ±8.1 | 72.07 ±2.0 | **77.00 ±3.0** | 71.27 ±10.3 | 10.3 ±5.4 | 68.79 ±5.6 | 59.92 ±5.1 |
| walk-m-e | 107.8 ±2.2 | **108.3 ±3.9** | **110.4 ±0.9** | **110.1 ±0.2** | 99.62 ±3.0 | **109.1 ±1.3** | **108.51 ±0.6** |
| lift-ph | 96.75 ±1.8 | 84.75 ±6.2 | 91.50 ±5.4 | 91.75 ±5.9 | 96.6 ±3.0 | **98.84 ±2.3** | 97.60 ±2.9 |
| lift-mh | 86.75 ±2.8 | **91.00 ±2.8** | **90.8 ±5.8** | 86.75 ±6.0 | 60.4 ±25.1 | **90.04 ±4.5** | 87.20 ±5.3 |
| can-ph | 74.50 ±6.8 | 68.00 ±9.1 | 62.00 ±10.9 | 69.67 ±5.9 | 63.0 ±20.3 | **76.40 ±3.7** | 74.8 ±2.4 |
| can-mh | 56.25 ±8.8 | 47.50 ±3.5 | 30.50 ±8.7 | 50.50 ±6.5 | 30.4 ±23.0 | 53.6 ±7.9 | **57.6 ±5.0** |
| Avg Std | 10.95 | 10.2 | 13.87 | 9.08 | 13.77 | 8.23 | **4.8** |

Table 1: Average normalized scores of all baselines on human-preference benchmarks from Kim et al. [28]. For the D4RL locomotion tasks "hop" corresponds to hopper, "m" to medium (training the data generating agent to 1/3 expert performance), "r" to replay buffer data, and "e" to data from the end of training. For the Robomimic tasks lift and can, "ph" corresponds to proficient human data and "mh" to multi-human data of differing optimality. The first four columns are taken from Kim et al. [28]. "reimpl." is our reimplementation of Markovian Reward with IQL. The "Avg Std" row shows the average standard deviation across all eight environments. We run five seeds and report the final performance at the end of training like Kostrikov et al. [30]. Bolded values are within 95% of the top performing method. Note that standard devaition values in the table were rounded for space. On some tasks IPL achieves higher performance earlier in training, which is not reflected above (See Appendix B). We find that IPL outperforms PT on many environments, and also performs similarly to our implementation of MR despite not training a reward function.

implementation of a MR (sixth column) performs far better than reported in Kim et al. [28], likely due to our careful tuning of $r_\theta$ and use of data-augmentations. Our method, IPL, achieves competitive performance across the board.

In general, IPL with IQL performs on-par or better than both our implementation of MR and PT in most datasets despite not learning a separate reward network. Specifically, IPL has the same performance or better performance than our MR implementation on six of eight tasks. More importantly, IPL does extremely well in comparison to Preference Transformer's reported results. On five of eight tasks IPL performs better than PT while having over 10 times fewer parameters, making IPL far more efficient. To be consistent with Kim et al. [28], we report results after a million training steps but performance for IPL often peaks earlier (see learning curves in the Appendix). For example, with early stopping IPL also outperforms PT on "hop-m-r". We posit that this is because the $Q$-function in IPL is tasked with both fitting the expert's preference model and optimal policy simultaneously, making both the policy and reward function non-stationary during training. In some datasets, this was more unstable.

IPL also has the lowest average standard-deviation across seeds, meaning it yields more consistent results than explicit reward methods. For standard two-phase preference-based RL algorithms, errors in the reward model are propagated to and exacerbated by the $Q$ function. IPL circumvents this problem by not explicitly learning the reward.

Finally, in Table 2, we consider various design decisions of IPL. Augmentations provide a strong boost in the robotics environment, but offer only minor improvements in locomotion. Removing regularization, however, is detrimental to performance. This is likely because without regularization, the implicit reward values can continue to increase, leading to exploding estimates of $Q$. Finally, we show that IPL is compatible with other offline RL algorithms by combining it with $\mathcal{X}$QL [19]. We find that with $\mathcal{X}$QL, IPL performs even better on some tasks, but worse on others. Finally, in Appendix B, we also show that IPL can be combined with online preference-based RL algorithms like PEBBLE [32].

### 4.3 How does IPL scale with Data?

Collecting preference comparisons is often viewed as the most expensive part of preference-based RL. To investigate how well IPL performs in data limited settings, we construct scripted preference datasets

| Dataset | No Aug | $\lambda = 0$ | IPL-XQL | IPL |
|---|---|---|---|---|
| hop-m-r | 70.46 ±6.7 | 10.41 ±2.26 | 80.4 ±2.13 | 73.57 ±6.67 |
| walk-m-r | 58.50 ±5.3 | 4.85 ±1.52 | 57.82 ±5.24 | 59.92 ±5.11 |
| lift-mh | 84.8 ±4.1 | 52.60 ±10.1 | 89.00 ±4.4 | 87.20 ±5.3 |
| can-mh | 53.2 ±5.8 | 13.8 ±5.7 | 59.0 ±5.0 | 57.6 ±5.00 |

Table 2: Ablations for IPL on the offline human-preference benchmark. We consider removing data augmentation, removing regularization $\lambda = 0$, and other offline RL algorithms ($\mathcal{X}$QL). Full results can be found in Appendix B.

| Preference Queries | | 500 | 1000 | 2000 | 4000 |
|---|---|---|---|---|---|
| Button Press | MR | **66.0** ±8.0 | 49.3 ±12.1 | 54.7 ±26.8 | 78.3 ±9.2 |
| | IPL | 53.3 ±8.5 | **60.1** ±12.8 | **70.2** ±2.5 | **90.2** ±6.5 |
| Drawer Open | MR | **65.9** ±9.9 | **87.2** ±5.2 | **89.7** ±6.4 | 94.6 ±3.9 |
| | IPL | 62.1 ±4.8 | 78.7 ±12.4 | 89.5 ±5.0 | **96.6** ±1.3 |
| Sweep Into | MR | 33.0 ±5.7 | 46.2 ±6.0 | **63.2** ±13.7 | **70.8** ±7.9 |
| | IPL | **34.5** ±2.3 | **48.2** ±7.2 | 58.8 ±7.4 | 65.9 ±6.7 |
| Plate Slide | MR | **54.6** ±5.3 | **57.2** ±4.5 | 23.9 ±18.8 | **55.2** ±3.0 |
| | IPL | 52.9 ±4.8 | 55.8 ±2.2 | **55.4** ±3.1 | 54.9 ±2.8 |
| Assembly | MR | 0.6 ±0.7 | 0.7 ±1.0 | 0.0 ±0.0 | 2.6 ±2.8 |
| | IPL | **0.9** ±0.6 | **1.5** ±1.5 | **1.7** ±1.9 | **5.5** ±5.2 |
| Avg Std | MR | 5.9 | **5.76** | 13.14 | 5.36 |
| | IPL | **4.2** | 7.22 | **3.98** | **4.5** |

Table 3: Results on five MetaWorld tasks at four different preference data scales. We run five seeds for each method, and take the highest average performance across seeds from the learning curves. More details can be found in Appendix B. IPL performs the same or better than IQL with a Markovian reward model on the majority of tasks and preference data scales without training a reward model.

of four different sizes for five tasks from the MetaWorld benchmark [56] used in prior preference-based RL works [32, 22]. We then train on the preference data $\mathcal{D}_p$ by setting $\mathcal{D}_o = \{(s, a, s') \in \mathcal{D}_p\}$ and use the same hyper-parameters for all environments and methods where applicable. Our results are summarized in Table 3. Again, IPL is a strong reward-free baseline. We find that at all data scales, IPL performs competitively to our implementation of MR (IQL with a learned Markovian reward) and consistently outperforms it in Button Press and Assembly. Increasing the amount of preference data generally improves performance across the board. However, as we generate queries uniformly at random some preference datasets may be easier to learn from than others, leading to deviations from this trend in some cases. As in the benchmark results in Table 1, IPL exhibits lower variance across seeds and tasks, in this case at three of four data scales.

### 4.4 How efficient is IPL?

One benefit of IPL over other preference-based RL methods is its parameter efficiency. By removing the reward network, IPL uses fewer parameters than other methods while achieving the same performance. In Table 4, we show the number of parameters for each method used in the last two sections. Preference Transformer uses over ten times more parameters than IPL, and the LSTM-based NMR model from Early et al. [15] uses nearly twice as many. When dealing with a limited compute or memory budget, this can be important. To exacerbate this effect, we consider an extremely parameter efficient version of IPL, denoted "IPL (64)" in Table 4, based on Advantage Weighted Actor Critic (AWAC) [39] which eliminates the second critic and value networks used in IQL [30] and uses a two-layer 64-dimensional MLP. We then compare this parameter-efficient IPL to MR with the same parameter budget which results in "MR (35)", a 35-dimensional MLP. Results are depicted on the left of Fig. 2. MR trained with a smaller network is unable to adequately fit the data, resulting in lower performance. Only after increasing the network size past that of IPL can MR begin to match performance.

Aside from parameter efficiency, IPL is also "hyper-parameter efficient". By removing the reward network, IPL removes a whole set of hyper-parameters associated with two phase preference based RL methods, like reward network architecture, learning rate, stopping criterion, and more. In the

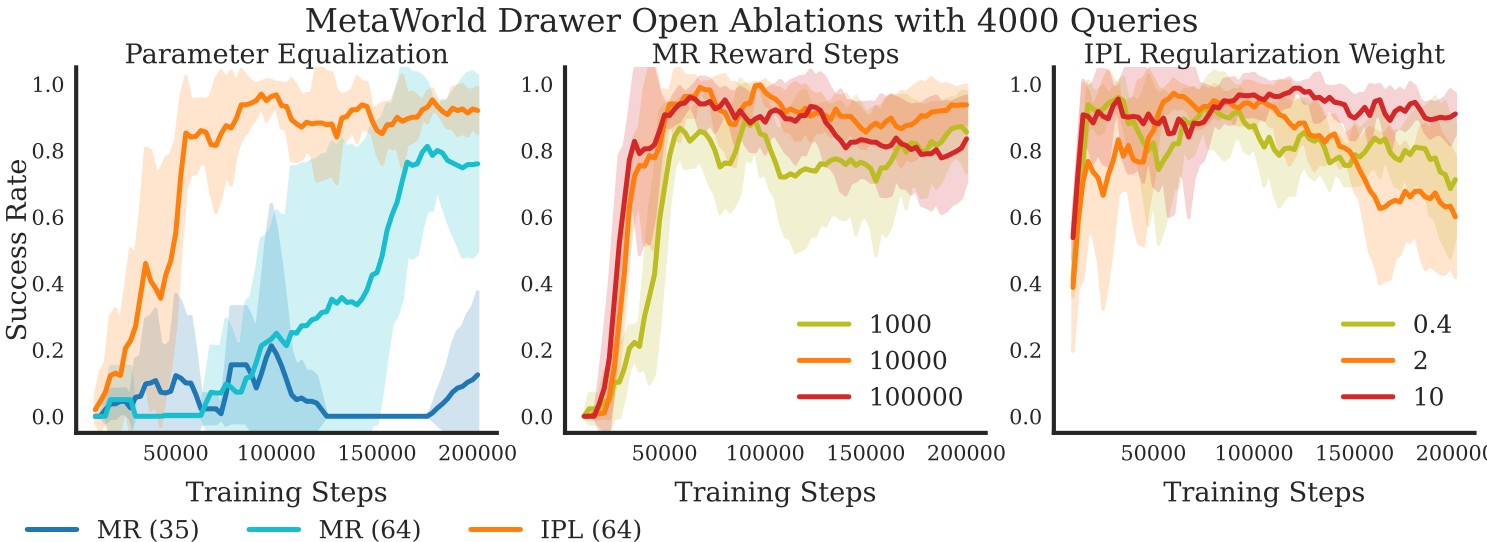

Figure 2: **Left:** Performance comparison with different parameter numbers. MR (35) has the same parameter budget as IPL (64). MR (64) has over twice as many. We see that with the same number of parameters as IPL, MR is unable to adequtely fit the data and performs poorly. **Middle:** MR when the reward function is trained for a varying number of steps – with too few the reward model under-fits, and with too many it over-fits, both leading to worse performance. **Right:** IPL with different regularization strengths. On the drawer open task, performance is largely unaffected. For more ablations, see the Appendix.

middle of Fig. 2 we show how the performance of MR is affected when the reward function is over or under fit. Choosing the correct number of steps to train the reward model usually requires collecting a validation set of preference data, which is costly to obtain. Instead of this, IPL only has a single regularization parameter, $\lambda$. The right side of Fig. 2 shows the sensitivity of IPL to $\lambda$. We find that in many cases, varying $\lambda$ has little effect on performance unless it is perturbed by a large amount.

## 5 Conclusion

**Summary.** We introduce Inverse Preference Learning, a novel algorithm for offline preference-based RL that avoids learning a reward function. Our key insight is to leverage the inverse soft-Bellman operator, which computes the mapping from $Q$-functions to rewards under a fixed policy. The IPL algorithm trains a $Q$-function to regress towards the optimal $Q^*$ while at the same time admitting implicit reward values that are consistent with an expert's preferences. Even though IPL does not require learning a separate reward network, on robotics benchmarks it attains competitive performance with preference-based RL baselines that use twice to ten-times the number of model parameters.

**Limitations and Future Work.** A number of future directions remain. Specifically, the implicit reward function and policy learned by IPL are both non-stationary during training, which sometimes causes learning to be more unstable than with a fixed reward function. This is a core limitation future work could address by better mixing

| Method | Params |
|---|---|
| PT | 2942218 |
| NMR | 508746 |
| MR | 348426 |
| IPL | 278537 |
| MR (64) | 34892 |
| IPL (64) | 14025 |
| MR (35) | 14012 |

Table 4: Parameter counts for different methods. The bottom three rows are for the limited parameter budget experiments in Section 4.4.

policy improvement and preference-matching steps to improve stability. More broadly, implicit reward preference-based RL methods are not limited to continuous control or binary feedback. Applying implicit reward techniques to other forms of feedback or extending IPL to language-based RLHF tasks remain exciting future directions.

## Acknowledgments and Disclosure of Funding

This work was supported by ONR, DARPA YFA, Ford, and NSF Awards #1941722 and #2218760. JH is supported by by the National Defense Science Engineering Graduate (NDSEG) Fellowship Program. We would additionally like to thank Div Garg and Chris Cundy for useful discussions.

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

# Appendix

## A    Theory

### A.1    Proofs

**Lemma 1** *For any fixed policy $\pi$ the inverse bellman operator $\mathcal{T}^\pi$ establishes a bijection between $r$ and $Q$. Moreover, for any $r$, $Q = (\mathcal{T}^\pi)^{-1}r$ is the unique fixed point of the Bellman operator $\mathcal{B}_r^\pi$. (Adapted from Garg et al. [18])*

*Proof.* Let $P^\pi$ be the stochastic transition matrix for the MDP corresponding to a fixed policy $\pi$. In vector form, the inverse bellman operator becomes $r = \mathcal{T}^\pi Q = (I - \gamma P^\pi)Q$. We can establish a bijection between $Q$ and $r$ by showing that $(I - \gamma P^\pi)$ is invertible. As $P^\pi$ defines a valid probability distribution over next stat-action pairs and $\gamma < 1$, we have that $||\gamma P^\pi|| < 1$. Thus, its Neumann series convergences, which implies the existence of $(I - \gamma P^\pi)^{-1}$. So, $Q = (I - \gamma P^\pi)^{-1}r$ and a bijection exists. Using this, we can also show a 1-1 mapping with the bellman operator under reward $r$. We have $Q = (I - \gamma P^\pi)^{-1}r = (\mathcal{T}^\pi)^{-1}r = \mathcal{B}_r^\pi Q$ at the fixed point of $\mathcal{B}_r^\pi$.

**Theorem 1** *Given an off-policy RL algorithm that convergences to the optimal policy $\pi_r^*$ for some reward function $r$ and regularizer $\psi$ such that Eq. (2) is strictly convex, IPL converges to $\pi_{r^*}^*$ corresponding to reward function $r^* = \arg\min_r \mathbb{E}_{\mathcal{D}_p}[D_{KL}(P_{r_E}||P_\theta)] + \lambda\psi(r)$.*

*Proof.* We prove this statement in the Tabular setting, first for algorithms that use policy improvement. Let $Q_t \in \mathbb{R}^{|S \times A|}$ and $\pi_t$ indicate the Q-function and policy after $t$ iterations. Let $Q_0 = 1/(1 - \gamma) \min_{S \times A} r(s, a)$. The inverse bellman operator tells us, in vector form, that $r = (I - \gamma P^\pi)Q$ where $P^\pi$ is the transition matrix. Let $r^* = \arg\min_r \mathbb{E}_{D_p}[y \log P_r + (1 - y)\log(1 - P_r)] + \lambda\psi(r)$, or the minimizer of the preference loss with regularizer $\psi$ such that we converge to a unique $r^*$.

At each step of IPL, we substitute the inverse bellman operator into the preference loss and optimize. Thus at convergence, $(I - \gamma P^{\pi_t})Q_t = r^*$ uniquely due to Lemma 1. Then, there are two cases based on the type of RL algorithm.

If our RL algorithm can directly estimate $\mathcal{B}_r^*$, then we are done. This is because we have assumed convergence, and thus $\mathcal{T}^* Q = r^*$. By the bijection established in Lemma 1, we have that $Q = (\mathcal{T}^*)^{-1}r^*$ which by also Lemma 1 is the unique fixed point of $\mathcal{B}_{r^*}^*$ which is $Q_{r^*}^*$. Thus, we have recovered the optimal $Q$ function for $r^*$ from which the optimal policy $\pi_{r^*}^*$ can be extracted.

If we have an RL algorithm that uses policy improvement, we consider multiple steps of IPL. If the RL algorithm guarantees convergence via policy improvement, then we use $\pi_t$ and $Q_t$. to obtain a new policy $\pi_{t+1}$. Using $\pi_{t+1}$ we can obtain the transition matrix $P^{\pi_{t+1}}$. Finally, we optimize the preference loss again using $P^{\pi_{t+1}}$ in the inverse Bellman operator to obtain $Q_{t+1}$. At convergence $(I - \gamma P^{\pi_{t+1}})Q_{t+1} = r^*$ holds. As $r^*$ is unique, $Q_t$ and $Q_{t+1}$ are both Q-functions for the reward function $r^*$, just under different policies. We know from the definition of policy improvement, that $Q^{\pi_{t+1}} \geq Q^{\pi_t}$ necessarily, and thus $Q_{t+1} \geq Q_t$ for any $t$. Convergence is possible as according to Lemma 1, $Q^* = (\mathcal{T}^*)^{-1}r$ is the a fixed point of $\mathcal{B}_r^*$.

**Proposition 1** *If $\phi(r) = r^2$, then IPL converges to the optimal policy corresponding to $r^* = \arg\min_r \mathbb{E}_{\mathcal{D}_p}[D_{KL}(P_{r_E}||P_\theta)] + \lambda\psi(r)$ in tabular settings.*

*Proof.* The preference-based loss function with L2 regularization can be viewed as L2 regularized logistic regression by writing the logits as a dot-product between a preference comparison vector $x$ comprised of $-\gamma^t, \gamma^t$ and 0 terms and a reward function $r$. The Hessian for this objective is then $X^T D X + \lambda I$     where $D_{ii} = \text{logistic}(x_i \top r)(1 - \text{logistic}(x_i \top r))$, which is positive definite. Thus the problem is strictly convex and $r^*$ is unique, so IPL converges to its optimal policy by Theorem 1. Note that to guarantee this we must regularize $r$ across the entire state-action space, analogous to regularizing the weight vector in logistic regression.

### A.2    Connections to DPO

Concurrent work, called Direct Preference Optimization (DPO) [45] also remove the need for explicit reward modeling for learning from preferences, but do so in the contextual bandits setting for language

models. Thus, DPO is limited to preference queries segment length 1 that *must start from the same state*. IPL is in fact, a more general version of DPO that does not have these restrictions. Specifically, IPL with $\mathcal{X}$QL recovers the same exact policy as DPO when applied to the contextual bandits setting.

Within the bandits setting, there is no "next-state" and $V^*(s')$ is removed, and the inverse bellman operator becomes just $Q(s, a) = r(s, a)$. The optimal $\mathcal{X}$QL policy is $\pi^* = \mu(a|s)e^{Q^*(s,a)}/Z(s)$ where $Z$ is the partition function. By rearranging, $\mathcal{T}^*Q = Q^*(s, a) = \log \frac{\pi(a|s)}{\mu(a|s)} + Z(s)$. We can plug this into the preference model induced by Q in Eq. (5). In the RLHF setting, the partition function cancels since we assume the context to be the same between preferences. This exactly results in the DPO algorithm, showing the DPO is in fact just an instantiation of IPL for contextual bandits.

### A.3   IPL with Rankings

IPL can easily be extended to rankings using a Plackett Luce model. Consider permutations $\tau$ over $K$ segments:

$$P_{r_E}(\tau) = \prod_{k=1}^{K} \left( \exp \sum_t r_E(s_t^{\tau_k}, a_t^{\tau_k}) \right) / d_k$$

where $d_k = \sum_{j=k}^{K} \exp \sum_t \gamma^t r_E(s_t^{\tau_j}, a_t^{\tau_j})$. Then, we make the same substitution using the inverse bellman operator giving us the permutation model $P_Q$ implied by the Q function, and run maximum likelihood estimation over the model with the preference loss $\mathcal{L}_p(Q) = \mathbb{E}_{\tau \sim \mathcal{D}_p} \left[ \log P_Q(\tau) \right] + \lambda \psi(r)$.

## B   Results

We divide this part of appendix into four different sections following the results section. Each section additionally provides hyper-parameters used for IPL in that section. The first section, setup, contains detailed information on the experimental setup and hyper-parameters used. The second section on benchmark results gives full learning curves for the experiments in Section 4.2. The third section provides full learning curves for the MetaWorld and Data-scaling experiments. The final Appendix section provides extended ablations.

### B.1   Setup

Here we provide the full algorithmic outline of IPL using Implicit Q-Learning [30] that mimics our implementation. While in practice the policy $\pi$ could be extracted at the end of training, we do it simultaneously as in [30] in order to construct learning curves.

---
**Algorithm 2:** IPL Algorithm (IQL Variant)

---
**Input :** $\mathcal{D}_p, \mathcal{D}_o, \lambda, \alpha$
**for** $i = 1, 2, 3, ...$ **do**
    Sample batches $B_p \sim \mathcal{D}_p, B_o \sim \mathcal{D}_o$
    Update $Q$: $\min_Q \mathbb{E}_{B_p}[\mathcal{L}_p(Q)] + \lambda \mathbb{E}_{B_p \cup B_o}[\mathcal{L}_r(Q)]$
    Update $V$: $\min_V \mathbb{E}_{B_p \cup B_o} \left[ |\tau - \mathbb{1}(Q(s, a) - V(s))| (Q(s, a) - V(s))^2 \right]$
    Update $\pi$: $\max_\pi \mathbb{E}_{\mathcal{D}_p \cup \mathcal{D}_o} [e^{\beta(Q(s,a) - V(s))} \log \pi(a|s)]$

---

Note that above we write the temperature parameter $\beta$ as done in IQL, instead of how it is usually done, using $\alpha$ in the denominator [19, 44].

When sampling batches of preference data $B_p \sim \mathcal{D}_p$, we take sub-samples of each segment $\sigma$ of length $s$. For a sampled data point $(\sigma^{(1)}, \sigma^{(2)}, y)$, we sample start $\sim \text{Unif}[0, 1, 2, ...k - s]$ and then let take $\sigma = s_{\text{start}}, a_{\text{start}}, ..., s_{\text{start}+s}$. We use the same start value across the entire batch.

Given that we run experiments using MLPs, all of our experiments were run on CPU compute resources. Each seed for each method requires one CPU core and 8 Gb of memory.

### B.2  Benchmark Results

Here we provide details for our experiments on the preference-based RL benchmark from Kim et al. [28]. We use the same hyperparameters as Kim et al. [28] and Kostrikov et al. [30] as shown in Table 5.

**Gym-Mujoco Locomotion.** Hopper and Walker2D agents are tasked with learning locomotion policies from datasets of varying qualities taken from the D4RL [17] benchmark. Preference datasets were constructed by Kim et al. [28] by uniformly sampling segments. Preference datasets for "medium" quality offline datasets contain 500 queries, while preference datasets for "expert" quality offline datasets contain 100 queries. Segment length $k = 100$ for all datasets, and were subsampled to length $s = 64$ by IPL and our MR (reimpl). Evaluation was preformed over 10 episodes every 5000 steps. Full learning curves are shown in Fig. 3.

**RoboMimic.** The RoboMimic datasets contain interaction data of two types: ph — proficient human and mh – multihuman. The multi-human data was collected from human demonstrators of mixed quality. The robot is tasked with learning how to lift a cube (lift) or pick and place a can (can). Preference datasets were again taken directly from Kim et al. [28]. Preference datasets of size 100 with segment lengths $k = 50$, randomly sub-sampled to length $s = 32$ were used for the ph datasets. Preference datasets of size 500 with segment lengths $k = 100$, randomly sub-sampled to length $s = 64$ were used for the mh datasets. Evaluation was performed over 25 episodes every 50000 steps. Full learning curves are shown in Fig. 4.

**Online Experiments.** We also test a combination of IPL with PEBBLE [32] on a few tasks in the MetaWorld benchmark. Results can be found in Fig. 5

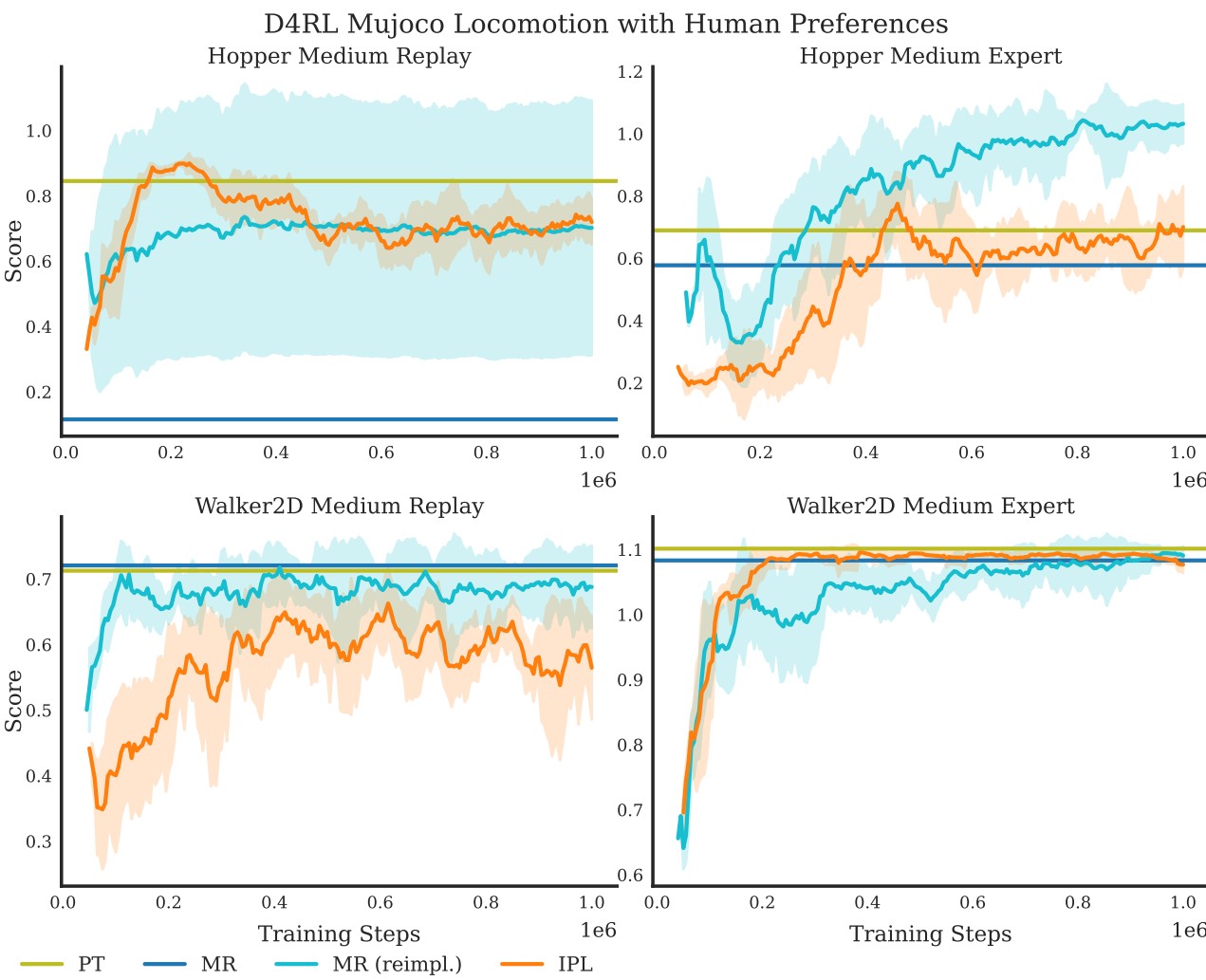

Figure 3: Full learning curves on the D4RL locomotion benchmark with human preferences.

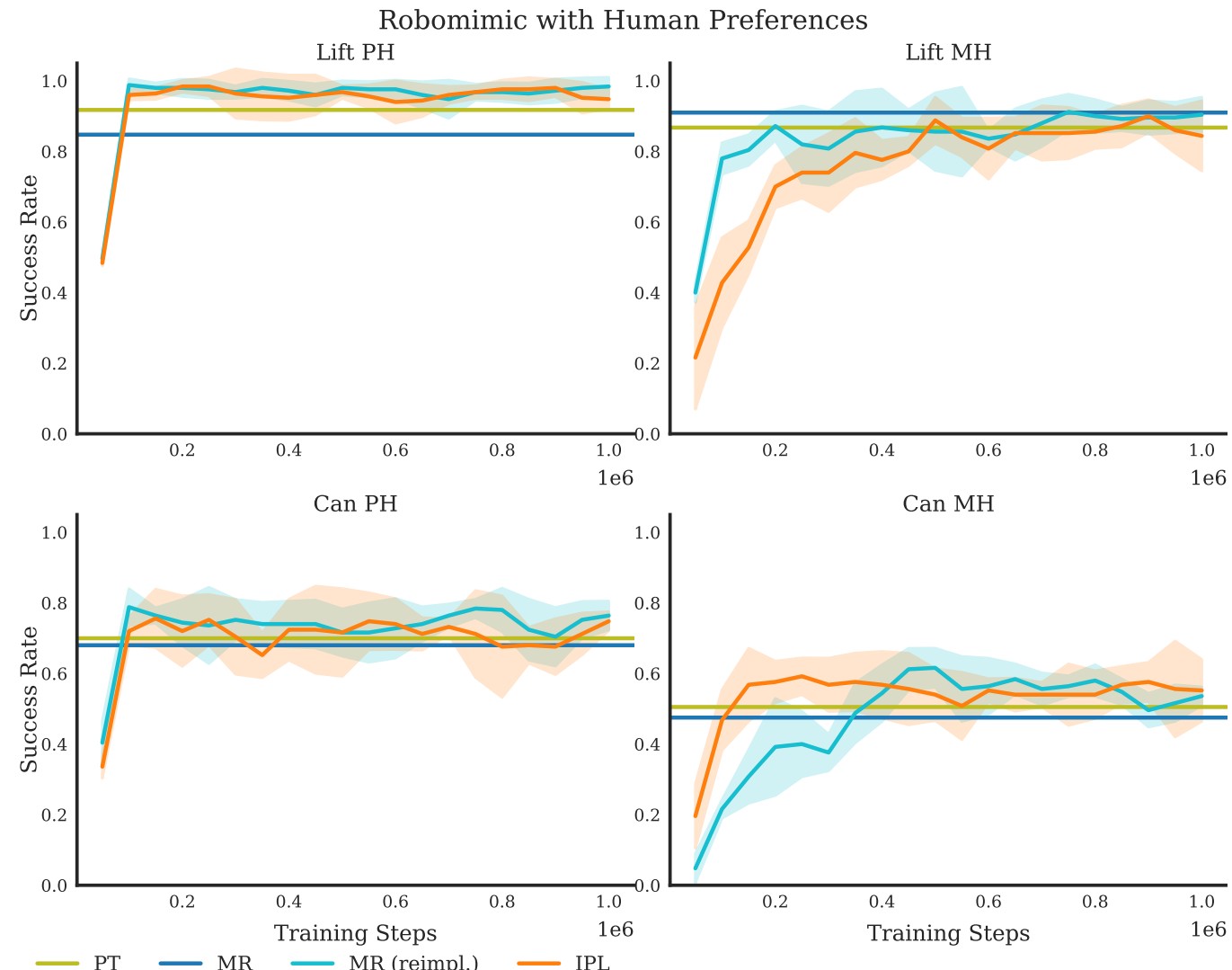

Figure 4: Full learning curves on the RoboMimic benchmark with human preferences.

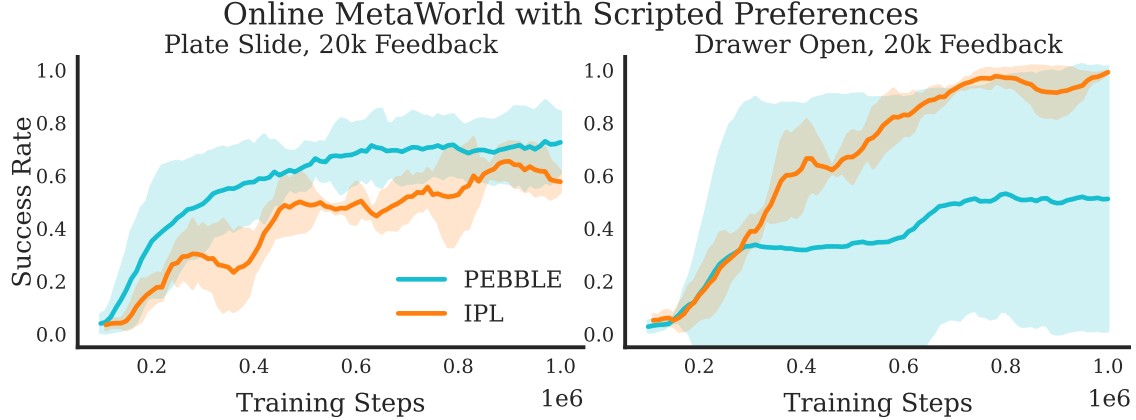

Figure 5: Online experiments for two of the most difficult MetaWorld environments. Each method (run on 3 seeds) gets feedback every 5000 steps for a total of 20k artificially queries. Notice that IPL has much lower variance across runs, particularly in Drawer.

| **Common Hyperparameters** | | | **MR Hyperparameters** | | |
| --- | --- | --- | --- | --- | --- |
| **Parameter** | **Locomotion** | **Robomimic** | **Parameter** | **Locomotion** | **Robomimic** |
| $Q, V, \pi$ Arch | 2x 256d | 2x 256d | $r_\theta$ Arch | 2x 256d | 2x 256d |
| Learning Rate | 0.0003 | 0.0003 | $r_\theta$ LR | 0.0003 | 0.0003 |
| Optimizer | Adam | Adam | $r_\theta$ Optimizer | Adam | Adam |
| $\beta$ | 3.0 | 0.5 | $r_\theta$ Steps | 20k | 20k |
| $\tau$ | 0.7 | 0.7 | | | |
| $\mathcal{D}_o$ Batch Size | 256 | 256 | | | |
| $\mathcal{D}_p$ Batch Size | 8 | 8 | | | |
| Training Steps | 1 Mil | 1 Mil | **IPL Hyperparameters** | | |
| $k$ | 100 | 100, 50 | **Parameter** | **Locomotion** | **Robomimic** |
| Subsample $s$ | 64 | 64, 32 | $\lambda$ | 0.5 | 4 |

Table 5: Hyperparameters used for the benchmark experiments. We can see that IPL has fewer hyperparameters. For the $X$QL experiments we use $\alpha = 2$ for locomotion and $\alpha = 5$ for Robomimic. We left all other parameters the same.

## B.3 Data Scaling Results

Experiments for data scaling were conducted on the MetaWorld benchmark from Yu et al. [56]. Offline datasets for five different MetaWorld tasks were constructed as follows: Collect 100 trajectories of expert data on the target task using the built in ground truth policies with the addition of Gaussian noise of standard deviation 1.0. Collect 100 trajectories of sub-optimal data by running the ground-truth policy for a different randomization of the target task with Gaussian noise 1.0. Collect 100 trajectories of even more sub-optimal data by running the ground truth policy *of a different task* with Gaussian noise standard deviation 1.0 in the target domain. Finally, collect 100 trajectories with uniform random actions. As MetaWorld episodes are 500 steps long, this results in 200,000 time-steps of data. We then construct preference datasets by uniformly sampling segments from the offline dataset and assigning labels $y$ according to $\sum_t r(s_t^{(1)}, a_t^{(1)}) > \sum_t r(s_t^{(2)}, a_t^{(2)})$ where $r$ is the ground truth reward provided by metaworld. We then train using only the data from $\mathcal{D}$ General architecture hyper-parameters were taken from Lee et al. [32], Hejna and Sadigh [22] which also use the MetaWorld benchmark, but for online preference-based RL. Full-hyper parameters are shown in Table 6. We run 20 evaluation episodes every 2500 steps. Full learning curves are shown in Fig. 6. When reporting values in Table 3, we choose the maximum point on the learning curves which average across five seeds. This provides results as if early stopping was given by an oracle, which is less optimistic than averaging the maximum of each seed as done in Mandlekar et al. [36].

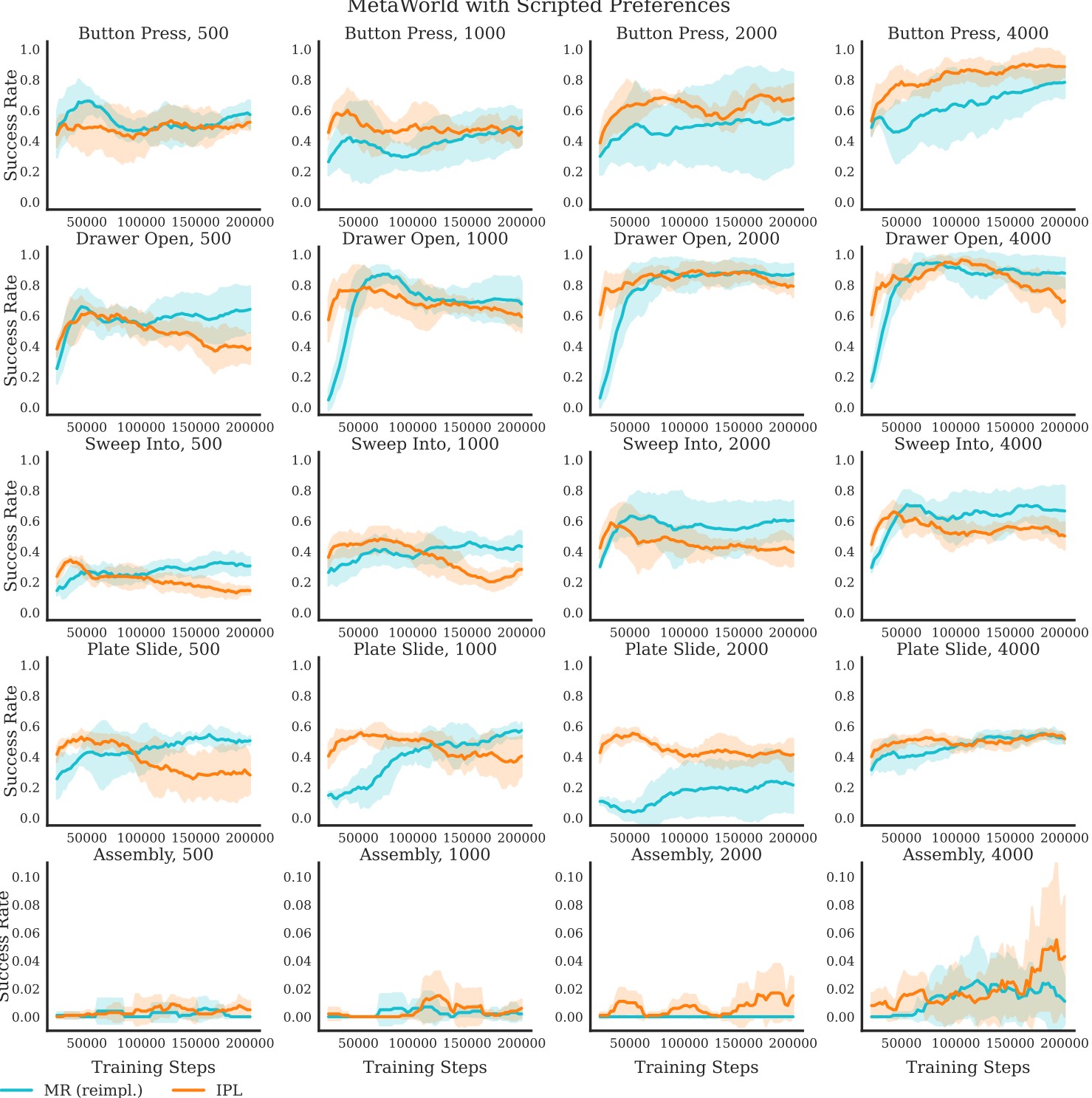

Figure 6: Full learning curves for the MetaWorld data scaling results with scripted preferences.

| Common Hyperparameters | | MR Hyperparameters | |
|---|---|---|---|
| **Parameter** | **Value** | **Parameter** | **Value** |
| $Q, V, \pi$ Arch | 3x 256d | $r_\theta$ Arch | 3x 256d |
| Learning Rate | 0.0003 | $r_\theta$ LR | 0.0003 |
| Optimizer | Adam | $r_\theta$ Optimizer | Adam |
| $\beta$ | 4.0 | $r_\theta$ Steps | 20k |
| $\tau$ | 0.7 | | |
| $\mathcal{D}_p$ Batch Size | 16 | **IPL Hyperparameters** | |
| Training Steps | 200k | **Parameter** | **Value** |
| $k$ | 25 | $\lambda$ | 0.5 |
| Subsample $s$ | 16 | | |

Table 6: Hyper-parameters used in the MetaWorld data scaling experiments.

## B.4 Ablations

In this section we provide additional ablations on both the benchmark datasets and MetaWorld datasets. We keep the hyperparameters the same, except for the parameter-efficient experiments.

**Benchmark IPL Ablations.** We include results of full ablations for IPL on the benchmark tasks in Table 7. We additionally provide comparisons between IPL and MR + IQL with and without data augmentation in Fig. 7.

| Dataset | No Aug | $\lambda = 0$ | IPL-XQL | IPL |
|---|---|---|---|---|
| hop-m-r | 70.46 ± 6.73 | 10.41 ± 2.26 | 80.4 ± 2.13 | 73.57 ± 6.67 |
| hop-m-e | 51.26 ± 17.46 | 52.81 ± 7.45 | 54.3 ± 12.33 | 74.52 ± 10.11 |
| walk-m-r | 58.50 ± 5.31 | 4.85 ± 1.52 | 57.82 ± 5.24 | 59.92 ± 5.11 |
| walk-m-e | 108.91 ± 0.18 | 58.77 ± 15.75 | 75.16 ± 23.40 | 108.51 ± 0.60 |
| lift-ph | 98.0 ± 2.53 | 85.2 ± 7.71 | 98.40 ± 2.59 | 97.60 ± 2.94 |
| lift-mh | 84.8 ± 4.11 | 52.60 ± 10.07 | 89.00 ± 4.37 | 87.20 ± 5.31 |
| can-ph | 68.6 ± 8.25 | 25.4 ± 5.25 | 68.6 ± 7.66 | 74.8 ± 2.40 |
| can-mh | 53.2 ± 5.8 | 13.8 ± 5.73 | 59.0 ± 5.0 | 57.6 ± 5.00 |

Table 7: Extended IPL ablation results.

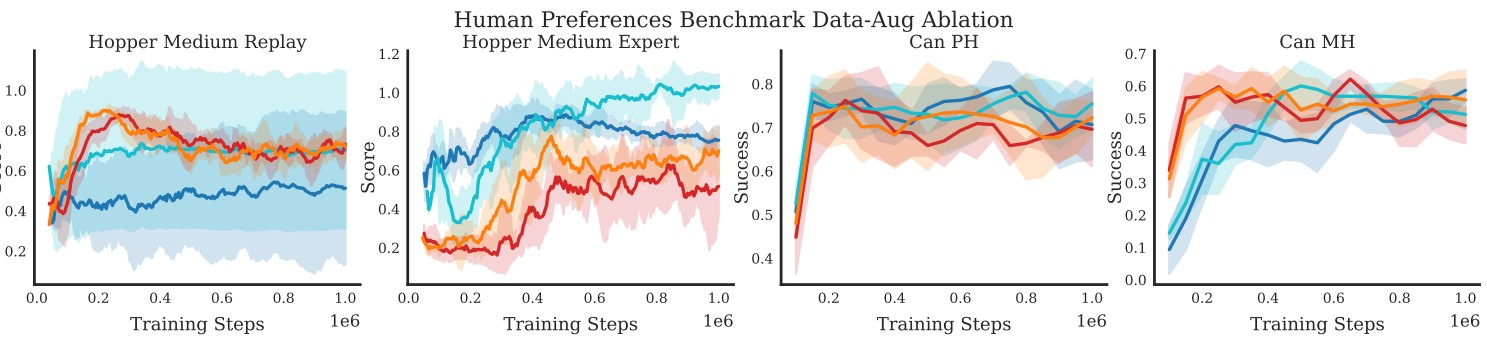

Figure 7: IPL and MR+IQL with and without data augmentation across 5 seeds. We see that data augmentation makes a large difference, especially for MR+IQL in the hopper environment, while its effects are less for the robomimic Can datasets.

**Hyper-parameter Sensitivity.** We run hyper-parameter sensitivty results for the human-preference benchmark datasets in Fig. 8. The top row depicts the sensitivity for IPL to the value of $\lambda$. The bottom row depicts the sensitivity of MR to the number of timesteps the reward function is trained for.

**Parameter Efficiency.** For the parameter-efficient experiments *only* we use an efficient version of IPL based on AWAC [39] to additionally remove the need for learning value network. AWAC uses

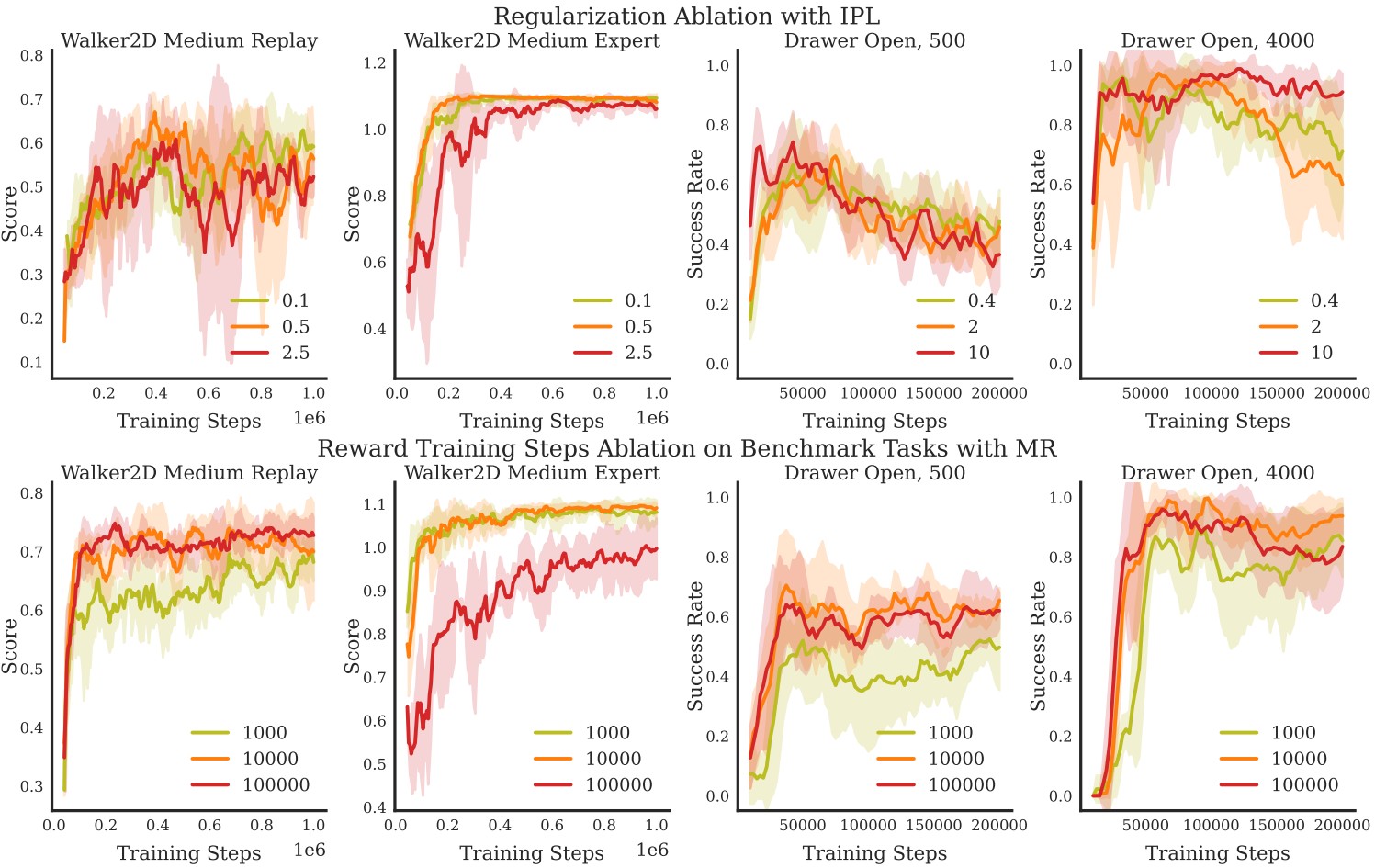

Figure 8: Ablations on regularization strength $\lambda$ for IPL (top row) and the number of reward steps for MR (bottom row). We see that IPL is relatively consistent across different values of $\lambda$. MR on the other hand, can vary greatly if the reward function under or over fits. In Walker2D Medium Replay and Drawer Open, 500, we see that it can easily under-fit. In Walker2D Medium Expert it easily over-fits.

a policy-evaluation, policy imporvement style approach. The outline of this variant is given below:

---
**Algorithm 3:** IPL Algorithm (AWAC Variant)

---
**Input :** $\mathcal{D}_p$, $\mathcal{D}_o$, $\lambda$, $\alpha$
**for** $i = 1, 2, 3, ...$ **do**
    Sample batches $B_p \sim \mathcal{D}_p$, $B_o \sim \mathcal{D}_o$
    Estimate $V$ as $Q(s, \pi(s))$
    Update $Q$: $\min_Q \mathbb{E}_{B_p}[\mathcal{L}_p(Q)] + \lambda \mathbb{E}_{B_p \cup B_o}[\mathcal{L}_r(Q)]$
    Update $\pi$: $\max_\pi \mathbb{E}_{\mathcal{D}_p \cup \mathcal{D}_o}[e^{\beta(Q(s,a)-Q(s,\pi(s)))} \log \pi(a|s)]$

---

For this version of IPL, we use $\lambda = 0.5$. All other hyper-parameters remain the same as in Table 8 except the architectures. For the parameter-efficiency experiments only we use MLPs consisting of two dense layers with either dimension 64 or dimension 35. Running MR with a two-layer MLP of dimension 35 has almost exactly the same number of parameters as IPL-AWAC with two-layer MLPs of dimension 64. We include full results for the parameter-efficiency experiments in Table 8. We find that on Drawer Open and Sweep Into, IPL outperforms both MR (64) and MR (35). In these environments, performance increases from MR (35) to MR (64) indicating that the expressiveness of the $Q$-function and policy are limiting performance. For the same budget, IPL is able to perform better. In Button Press, the simplest task, we find that MR (64) actually over-fits more than MR (35) and MR (64) ends up performing worse. In Plate Slide, all methods perform similarly independent of parameter count. We omit Assembly because of its low success rate at all data scales.

| Preference Queries | | 500 | 1000 | 2000 | 4000 |
|---|---|---|---|---|---|
| | MR (35) | **73.9** $_{\pm 8.9}$ | **86.8** $_{\pm 8.2}$ | **89.9** $_{\pm 14.4}$ | **99.0** $_{\pm 1.0}$ |
| Button Press | MR (64) | 54.2 $_{\pm 16.1}$ | 42.6 $_{\pm 33.0}$ | 67.1 $_{\pm 14.9}$ | 43.4 $_{\pm 7.4}$ |
| | IPL (64) | 65.8 $_{\pm 13.3}$ | 79.8 $_{\pm 18.1}$ | 80.0 $_{\pm 17.3}$ | **95.8** $_{\pm 5.2}$ |
| | MR (35) | 13.4 $_{\pm 13.9}$ | 12.6 $_{\pm 21.9}$ | 15.5 $_{\pm 20.1}$ | 18.4 $_{\pm 25.6}$ |
| Drawer Open | MR (64) | 13.4 $_{\pm 19.0}$ | 57.1 $_{\pm 31.2}$ | 54.5 $_{\pm 31.7}$ | 78.8 $_{\pm 12.2}$ |
| | IPL (64) | **89.8** $_{\pm 11.3}$ | **93.2** $_{\pm 2.5}$ | **99.5** $_{\pm 0.9}$ | **95.5** $_{\pm 3.7}$ |
| | MR (35) | 35.1 $_{\pm 8.9}$ | 42.4 $_{\pm 9.9}$ | 45.9 $_{\pm 9.6}$ | 35.9 $_{\pm 4.1}$ |
| Sweep Into | MR (64) | 31.1 $_{\pm 6.4}$ | 55.8 $_{\pm 5.9}$ | 49.6 $_{\pm 10.3}$ | 56.4 $_{\pm 10.3}$ |
| | IPL (64) | **41.1** $_{\pm 14.2}$ | **63.9** $_{\pm 8.0}$ | **65.0** $_{\pm 12.0}$ | **63.9** $_{\pm 11.8}$ |
| | MR (35) | **55.2** $_{\pm 6.1}$ | **51.1** $_{\pm 4.4}$ | **53.0** $_{\pm 2.0}$ | **48.9** $_{\pm 3.3}$ |
| Plate Slide | MR (64) | 46.6 $_{\pm 21.9}$ | **50.8** $_{\pm 0.6}$ | 47.0 $_{\pm 2.5}$ | **48.5** $_{\pm 4.6}$ |
| | IPL (64) | **54.9** $_{\pm 3.2}$ | **49.4** $_{\pm 1.6}$ | 45.2 $_{\pm 9.0}$ | **48.8** $_{\pm 4.9}$ |

Table 8: Performance of different methods on the MetaWorld tasks under a limited parameter budget. MR (35) and IPL (64) have the same number of parameters. The Assembly task is ommited due to low success rate. On Button Press, fewer parameters appears to perform better as, due to the simplicity of the task, its easier for the bigger models to overfit. On Drawer Open and Sweep Into, we see consistent gains from increasing the number of parameters in the network, and IPL performs best overall. On the Plate Slide task, all methods at different parameter scales perform similarly.

