# OpenReview forum: "Inverse Preference Learning: Preference-based RL without a Reward Function"
_NeurIPS.cc/2023/Conference — NeurIPS 2023 poster_

### Official Review · Reviewer_HK3n · 2023-06-14

**Soundness:** 3 good
**Presentation:** 4 excellent
**Contribution:** 2 fair
**Rating:** 5
**Confidence:** 4

**Summary:**

The paper proposes a way to learn from preferences in the offline setting without learning a reward model. The main insight is that reward and Q functions are interchangeable and the policy learning using preference can be formulated directly as a function of Q which represents the reward function implicitly. This allows for increased learning performance without requiring to train a seperate reward function when doing RL with preferences.

**Strengths:**

Strengths:

1. The paper presents a new way to do policy learning from preference data without requiring learning an intermediate reward function.
2. The experiments demonstrate that their method IPL outperforms baselines that learn from offline preferences in a number of prior tasks that include simulated MuJoCo locomotion tasks and manipulation tasks.

**Weaknesses:**

1. Missing theoretical underpinnings: A number of questions come up with the proposed method that should be addressed:
    1. Is the learned policy optimal for the implicit Q function?

        Equation 6 is the probability that one trajectory is preferred by other under the policy $\pi$ with the reward function $T^\pi Q$ . Replacing $T^\pi Q$  with $T^* Q  = Q(s,a)-\gamma E[V^*(s')]$ , it is claimed that the preference learning objective will fit the optimal value function. There seems to be jump to this claim without a proof. In my opinion, it is important to show how minimizing Equation after line 201 leads to soft optimal $Q^*$ with ground truth reward function $r^E.$

    2. Large regularization weight

        A large regularization weight (1 and sometimes even larger than the main preference loss e.g in fig 2) is used in the work which makes it more confusing as to what combined objective is being optimized. While the intuition given by the authors makes sense, it might be great to show how this regularization does not change the stationary point of $Q^*$.

2. Clarification on novelty:
It is mentioned multiple times in the paper that:
”Line 163: Our key insight is that Q-function learned by off-policy RL algorithm in face encodes the same information as the reward function”

This is not a new insight. It has been presented in multiple prior works [1,2,3] where a change of variables for reward is performed to remove the intermediate step of reward learning for imitation learn and the mapping from reward to Q has been studied theoretically in the work of [4] and more recently [5]. It seems the novelty is to add the preference loss to the IQ-Learn loss function.

3. Prior literature on reward learning:

    Three baselines are used to compare the method IPL - MR, LSTM and PT. A number of other prior works exist which aim to do reward learning from preferences [6,7,8] and would be interesting to compare experimentally to and discuss the relationship in the paper.

4. Gap between proposed method and practical method: The paper discusses learning an optimal value function using the linex loss of XQL[9] but in the experiment section the value update is performed using IQL. Preference loss uses soft-optimal-Q whereas the policy update step in IQL updates for optimal Q. There seems to be a disconnect between the proposed method and practical method which might be important to address.
5. Experiments:
a. How is the preference dataset constructed? Elaborating that in the main paper can increase the understanding considerably.

    b. Limited experiments: Only 1 baseline is compared against in the meta-world task and 4 simulated domains in total are considered in the locomotion and robomimic tasks. With limited theoretical understanding of the method, it might be important to establish the method empirically with previous baselines in order to show its merits.


[1] Kostrikov, Ilya, Ofir Nachum, and Jonathan Tompson. "Imitation learning via off-policy distribution matching." *arXiv preprint arXiv:1912.05032* (2019).

[2] Nachum, Ofir, and Bo Dai. "Reinforcement learning via fenchel-rockafellar duality." *arXiv preprint arXiv:2001.01866* (2020).

[3] Ma, Yecheng Jason, et al. "Smodice: Versatile offline imitation learning via state occupancy matching." *arXiv e-prints* (2022): arXiv-2202.

[4]Garg, Divyansh, et al. "Iq-learn: Inverse soft-q learning for imitation." *Advances in Neural Information Processing Systems* 34 (2021): 4028-4039.

[5] Sikchi, Harshit, Amy Zhang, and Scott Niekum. "Imitation from Arbitrary Experience: A Dual Unification of Reinforcement and Imitation Learning Methods." *arXiv preprint arXiv:2302.08560* (2023).

[6]: Brown, Daniel, et al. "Safe imitation learning via fast bayesian reward inference from preferences." *International Conference on Machine Learning*. PMLR, 2020.

[7]: Chen, Letian, Rohan Paleja, and Matthew Gombolay. "Learning from suboptimal demonstration via self-supervised reward regression." *Conference on robot learning*. PMLR, 2021.

[8]: Sikchi, Harshit, et al. "A ranking game for imitation learning." *arXiv preprint arXiv:2202.03481* (2022).

[9]: Garg, Divyansh, et al. "Extreme Q-Learning: MaxEnt RL without Entropy." *arXiv preprint arXiv:2301.02328* (2023).

**Questions:**

1. Gradient through target Q: When implementing the regularization on  $T^* Q  = Q(s,a)-\gamma E[V^*(s')]$ it seems that only Q(s,a) is updated which means that reward is not being made to be zero centered and rather Q is made to be zero-centered.
2. XQL vs IQL: The main paper discusses XQL and the experiments use IQL. Is there a reason behind this disconnect? Are there XQL experiments to validate?

Update: Thanks to the authors for responding to a number of my questions and providing additional baselines. I have updated my score (4->5) and will reiterate on my score upon further discussion with reviewers. I hope to see the clarification on novelty and the additional baselines in the updated paper.

**Limitations:**

Yes.

---

> ### Author Rebuttal · Authors · 2023-08-10
>
> We would like to thank the reviewer for their time and effort in reviewing our work and have made a number of changes as a result.
>
> **Theoretical Underpinnings**
>
> The reviewer raised a number of questions regarding the theoretical underpinnings of IPL, namely its optimality and regularization. We provide a proof of IPL’s convergence in our central response due to space constraints. It depends on a general proof of the bijection, which was included in our response to Reviewer ggMH, and shows that IPL returns the optimal policy for the regularized expert reward.
>
> We hope the proof addresses the following concerns of the reviewer:
>
> 1. IPL is formally characterized, increasing understanding across the board.
> 2. We have shown that IPL optimizes for the regularized reward $r^*$
> 3. We have shown that IPL converges to the optimal policy for this reward.
>
> The reviewer also asked if the reward function we optimize is $r_E$ or not. $r^*$ is not equal to $r_E$. However, we would like to note that most preference based RL works do not optimize for $r_E$. For example, PEBBLE [2] bounds the reward values with a Tanh network in practice, which as a multiplicative clamp does not preserve $r_E$. Preference Transformer [1] normalizes reward by scaling it via the max and min episode, which is another multiplicative transform that does not preserve $r_E$. Like all these methods, the policy recovered by IPL is not exactly that of $r_E$, though they are likely similar. We will update the text to make this more clear, along with our proofs.
>
> The reviewer also asked about large regularization coefficients. Empirically we found that the stronger the regularization, the more bounded the $Q$-function becomes. Practically we found that in some domains a larger regularization weight was useful for some tasks – and this might have to do with the data. With higher quality data, there might not be too much difference between the $Q$ values for a good and bad segment. However, There is also an intuitive argument – that humans have relatively smooth preferences.  For example, humans would probably attribute reward smoothly across a video clip instead of attributing it all to a single frame.
>
> **Clarification of Novelty**
>
> Thank you for bringing this up. We apologize if the contribution statement was not clear. We have now made it clear that our contribution is the application of the inverse bellman operator to the Preference-based RL, not the inverse-bellman operator itself.
>
> **Prior Literature and Other Baselines**
>
> We will make sure to include citations for these other, relevant works. However, each of these works seeks to solve is slightly different from the Preference-based RL problem that we deal with. To our knowledge, Preference Transformer is state of the art in all of our benchmark tasks, which is why we selected it as a strong baseline.
>
> “Safe Imitation Learning Via Fast Bayesian Reward Inference From Preferences” from Brown et al. is largely designed for image-based experiments (our benchmarks are state), uses a number of self-supervised losses, and uses entire demonstrations instead of trajectory snippets. However, most critically this method requires hundreds of online MCMC rollouts during training, making it inapplicable to our offline benchmarks.
>
> “Learning from Suboptimal Demonstration via Self-Supervised Reward Regression” by Chen et al. is not a preference-based RL method. Instead, it is an inverse-RL method that improves upon D-REX to generate preference data for improving the policy. They assume access to a set of demonstrations, and no rankings. Their method addresses a different complementary challenge than IPL (inverse RL) by generating preference data. In this way, it is complementary to IPL and could be used to generate more data. It also requires online samples (unlike IPL) in order to generate extra data from noised policies.
>
> “A Ranking Game For Imitation Learning” by Harshit et al. is also an online inverse RL method, which can leverage some preference data. It is designed to be used online with access to a set of demonstrations, unlike the methods we test against.
>
> While these baselines are highly relevant, they aren’t exactly designed for the same setup we are concerned with. Specifically, most either address the inverse-RL problem and require online samples.
>
> **Gap Between Proposed and Practical Method**
>
> See response to reviewer ggMH under Soft-Q vs Standard.
>
> **Experiments**
>
> 1. *How is the preference dataset constructed?* Our datasets are taken from PT [1]. They are constructed by sampling segments from existing offline benchmarks (D4RL, robosuite), and asking a human to choose their preferred segment.
>
> 2.  *Limited Experiments* We compare all methods across eight standard benchmark tasks. The MetaWorld experiments were designed to show that across even more tasks and datasets scales IPL matches the performance of the best baseline that uses an explicit reward model (MR+IQL). This addresses the central motivation of our method – we can remove the reward function and still perform as well as the best baseline.
>
> If the reviewer feels strongly about the MetaWorld tasks, we are happy to run them with Preference Transformer for the final paper, but are compute limited and do not have the bandwidth to finish them immediately as this would require 5 tasks * 4 data scales * 5 seeds = 100 large transformer reward models, and as an academic lab we don’t have the resources available to do that on a short time-scale.
>
> **Question 1: Grad in Regularization**
>
> We apply a “stop gradient” on the target Q or V. The regularization $r^2 = (TQ)^2 = (Q(s,a) - \gamma V(s’))^2$ thus encourages $Q(s,a) = \gamma V(s’)$, which would imply that the implicit reward is zero. The $Q$-function is encouraged to be centered at the value.
>
> **Question 2: XQL vs IQL**
>
> See response to reviewer ggMH under Soft-Q vs Standard.
>
> [1] Kim et al.. Preference transformer. ICLR 2023.
>
> [2] Lee et al., PEBBLE. ICLR 2021.

---

> > ### Comment · Reviewer_HK3n · 2023-08-12
> > **Reviewer response**
> >
> > Thanks for the response. I have some follow-up comments
> >
> > > Theoretical underpinnings
> >
> > Thanks for the response on clarifying the convergence. My question was actually meant to ask a related question to what you described but still seems to be missing: How is $r^*$ related to the equivalence class of expert reward functions? What is the potential suboptimality incurred due to this regularization?
> >
> >
> > > Preference model learns Q using soft-Q based preference model whereas IQL updates does not use soft-Q model.
> >
> > In the experiments, Q is learned via soft updates whereas V uses hard updates using IQL. I would be curious to see how XQL fares in the paper/discussion since that seems to be a theoretically principled algorithm. The experiments for XQL shouldn't be memory intensive and quite fast to run.
> >
> > > "we have updated the method section with a new expanded derivation of IPL under any off-policy RL algorithm that works via policy evaluation and policy improvement steps"
> >
> > Under a general off-policy RL method, I dont see how a closed form solution for optimal value in terms of Q can exist. Specifically, how do you go from equation after line 172 to equation after line 194? If it requires learning another policy that serves as a maximizer of Q function, that seems to defeat the point since we have just replaced the reward function for a policy network.
> >
> > > Proof that IPL Converges to the Optimal Policy corresponding to the regularized expert reward?
> >
> > I am not sure I see why $r^*$ should be unique. Consider a 4 state ($s\_1,s\_2,s\_3,s\_4)$ MDP with deterministic transitions. Let the dataset have uniform probability over these states:
> > $s\_1$ -> $s\_2$ -> $s\_3$
> >                              -> $s\_4$
> > and suppose my preferences are $s\_1$ -> $s\_2$ -> $s\_3$ > $s\_1$ -> $s\_2$ -> $s\_4$ and one of the learned $r^*$ is (under the regularized preference loss):
> > r($s\_1$ )  = 1
> > r($s\_2$ )  = 2
> > r($s\_3$ )  = 3
> > r($s\_4$ )  = 0
> > Then another possible solution that achieves the same optima is:
> >
> > r($s\_1$ )  = 1
> > r($s\_2$ )  = 3
> > r($s\_3$ )  = 2
> > r($s\_4$ )  = 0
> >
> > Is it possible to prove uniqueness of reward function?
> >
> >
> > > Significant writing changes to clarify contributions
> >
> > Thanks for acknowledging this, and I believe on reading the paper again that significant writing changes need to be made to make sure the contributions are clarified.
> >
> > > "“Safe Imitation Learning Via Fast Bayesian Reward Inference From Preferences” from Brown et al. is largely designed for image-based experiments (our benchmarks are state), uses a number of self-supervised losses, and uses entire demonstrations instead of trajectory snippets. However, most critically this method requires hundreds of online MCMC rollouts during training, making it inapplicable to our offline benchmarks."
> >
> > I believe this work does use trajectory snippets instead of entire demonstrations as the authors suggest. MCMC rollouts are made on a linear reward function making it efficient and fast. Prior work, "Extrapolating Beyond Suboptimal Demonstrations via Inverse Reinforcement Learning from Observations" even used a pointwise estimate for reward without MCMC. Both of these baselines can work in the exact setup described here by replacing online RL with offline RL.

---

> > > ### Author Response · Authors · 2023-08-14
> > > **Response & Clarifications**
> > >
> > > Thank you for engaging, we appreciate your response and are seeking to clarify any misconceptions and improve our paper.
> > >
> > > > Equivalence class of reward
> > >
> > > We don’t believe so. We think formally investigating this would take a significant amount of time warranting its own work. We would like to point out that many works in preference-based RL leverage other similar techniques for regularizing the reward function without analysis like PEBBLE and PT.
> > >
> > > > Experiments for XQL
> > >
> > > We ran experiments with XQL (amounts to changing theV loss in IQL) with alpha = 2 for locomotion and alpha = 5 for robomimic chosen based on the XQL publication for Hopper, Walker2d, and Franka Kitchen. We did no additional tuning. Our results show the avg performance (3 seeds) at the end of training, with the max avg in parenthesis. NOTE: Please refer to updated table below for results with tuning -- we increased the value of alpha for locomotion.
> > >
> > > | Dataset | IPL+IQL       | IPL+XQL     |
> > > |---------|---------------|-------------|
> > > | h-m-r   | 73.6 (90)     | 1.7 (11.7)  |
> > > | h-m-e   | 74.5 (77.5)   | 20.3 (33.3) |
> > > | w-m-r   | 59.9 (66.3)   | 2.4 (15.0)  |
> > > | w-m-e   | 108.5 (109.5) | 1.9 (76.3)  |
> > >
> > > Consistent with results in [1], we see that XQL is unstable due to its objective. Recent works show that choosing different losses on V (like IQL) result in different regularizers that perform better [2].
> > >
> > > > Method Section New Derivation:
> > >
> > > To be more general, we wanted to extend our derivation of IPL to the standard approach for off-policy RL in continuous spaces: policy evaluation using $B^\pi$, where $B$ is the bellman operator, followed by policy improvement (usually greedy approximated by max Q, where there might not be a closed form). We thought the policy evaluation-improvement paradigm would be more general, as it is used to prove convergence for popular off-policy methods like SAC. In the context of IPL, this would mean using the inverse operator $T^\pi$, and repeatedly improving $\pi$.  In fact, in our data-limited experiments we used AWAC, an off-policy algorithm that operates in this way. For algorithms that directly approximate $B^*$ (XQL), we could correspondingly use $T^*$ to combine policy evaluation with policy improvement.
> > >
> > > When using AWAC the equation after line 194 would use $V^\pi(s’) = E_{a' \sim \pi(\cdot | s')} [Q(s',a')]$, and SAC would use $V^\pi(s’) = E_{a' \sim \pi(\cdot | s’)}[Q(s’,a') - \log \pi(a' | s’)]$. The objective function (Eq 3 in the paper) would consequently also change to match that of the chosen RL algorithm. For these updates we are not using the closed form. At the end of the day we need a policy, and all offline RL algorithms need potential function for improvement. IPL with SAC and AWAC both use just an actor and a critic, the same number of networks as SAC and AWAC with known reward. IPL with any algorithm uses the same number of networks as that algorithm. We do not understand why the policy network would be redundant as the reviewer states, or how we could reduce the number of networks below what is required of the base algorithm.
> > >
> > > We are open to sticking to the publication's current presentation of the method using $T^*$ if the reviewers think it is more clear.
> > >
> > > > IPL Convergence Proof
> > >
> > > We apologize for not making our assumptions clear in the proof, and will update to make this more clear.
> > >
> > > We agree that the example provided by the reviewer is under-determined. There isn’t sufficient data to specify $r_E$ up to a constant equivalence $r_E + c$ even with standard BCE. Both solutions provided by the reviewer achieve the same un-regularized preference loss. We do not argue that this problem is fixed with regularization – it remains ambiguous.
> > >
> > > Our proof assumes that we have sufficient preference data to fit $r_E + c$ with the standard BCE preference loss, and a regularizer that removes ambiguity over $c$. We have added an addendum to our response to explicitly state this: Assume data and regularizer $\psi$ s.t. $r^*$ is unique.
> > >
> > > For ex, consider a simple preference optimization with $r_1$, and $r_2$. Let the sigmoid function be $g$. The BCE loss is $-y \log g(r_1 - r_2) - (1-y) \log (1 - g(r_1 - r_2))$. Taking the gradient with respect to $r_1$ and setting it to zero results in the condition $y = g(r_1 - r_2)$. Taking the gradient of the loss wrt $r_2$ results in the same condition! Thus, the system is under-determined since we can add any constant to $r_1$ and $r_2$.
> > >
> > > If we add regularization $\lambda r_1^2 + \lambda r_2^2$ to the loss and solve for the point where the gradients both equal zero we arrive at  the condition $r_1 = -r_2$. For label y = 0 or 1 we can determine the system. If we didn’t have regularization the reward function could shift by $c$, making it hard to argue about policy improvement using $Q$.
> > >
> > > [1] Sikch et al.  Dual RL: Unification and New Methods 2023
> > >
> > > [2] Offline RL with No OOD Actions. Haoran Xu et al. ICLR 2023

---

> > > > ### Author Response · Authors · 2023-08-16
> > > > **Added additional experiments**
> > > >
> > > > Dear reviewer,
> > > >
> > > > We also wished to respond to your questions about experiments, but needed more time to complete them. We wanted to let you know that we have, and correspondingly introduce experiments on the preference benchmark across three new baselines / ablations:
> > > >
> > > > 1. IPL + XQL, for which we have tuned the value of alpha, and now achieves comparable performance to IPL + IQL, and better in some environments. Note that on walker2d-expert performance crashes halfway through training due to instability which is not reflected in the Table but will be in our final learning curves. Note that we kept the evaluation performance at the end of training like all other methods, instead of taking the maximum over training like the XQL authors.
> > > > 2. IPL Without Reg: We remove the regularization from our method to show its importance to performance empirically
> > > > 3. BREX (Safe imitation Leraning via Fast Bayesian Reward Inference from Preferences): As the reviewer requested we ran the BREX baseline. We did so by using the BREX author's original code to learn the reward function, and then optimizing the reward function using IQL for consistency with other methods. PPO, as used in the original BREX paper, was designed for online learning. We use their hyper-parameters. Some random reward initializations did not converge, and we restarted these to give the baseline a better chance. Since our experiments were in state, we used random gaussian projections of the state as the features.
> > > > 4. TREX: the reviewer mentions TREX, which we agree is important to acknowledge as highly relevant prior work. The TREX algorithm with our preference is actually equivalent to the MR+IQL baseline we already include. TREX uses partial trajectories with data augmentation (we use this) to learn a reward function, and then optimizes the reward function downstream. While the original TREX code uses PPO, we used IQL to better fit the offline setting. To our knowledge, TREX was originally designed for ranked demonstrations in imitation learning, but since we do not have ranked demonstrations, we use it with the same preference datasets.
> > > >
> > > > We would like to again emphasize that of all the methods, IPL is the only one that does not learn a reward network. Our goal remains to show that without learning a reward function, IPL can attain the same performance as other methods. BREX learns the simplest reward network, and correspondingly appears to have worse performance in environments where more complex reward features are required.
> > > >
> > > > Our additional ablations on IPL show that a) IPL is robust to the choice of underlying RL algorithm (XQL vs. IQL), and b) that regularization is important to performance.
> > > >
> > > > |                     | MR                | LSTM              | PT                | MR (reimpl)       | BREX              | IPL Without Reg.  | IPL+XQL           | IPL+IQL           |
> > > > |---------------------|-------------------|-------------------|-------------------|-------------------|-------------------|-------------------|-------------------|-------------------|
> > > > | hopper-med-replay   | 11.56 $\pm$ 30.27 | 57.88 $\pm$ 40.63 | 84.54 $\pm$ 4.07  | 70.20 $\pm$ 35.0  | 62.0 $\pm$ 20.29  | 10.41 $\pm$ 2.26  | 80.4 $\pm$ 2.13   | 73.57 $\pm$ 6.67  |
> > > > | hopper-med-expert   | 57.75 $\pm$ 23.70 | 38.63 $\pm$ 35.58 | 68.96 $\pm$ 33.86 | 102.97 $\pm$ 5.55 | 85.1 $\pm$ 7.97   | 52.81 $\pm$ 7.45  | 54.3 $\pm$ 12.33  | 74.52 $\pm$ 10.11 |
> > > > | walker2d-med-replay | 72.07 $\pm$ 1.96  | 77.00 $\pm$ 3.03  | 71.27 $\pm$ 10.30 | 68.79 $\pm$ 5.64  | 10.30 $\pm$ 5.44  | 4.85 $\pm$ 1.52   | 57.82 $\pm$ 5.24  | 59.92 $\pm$ 5.11  |
> > > > | walker2d-med-expert | 108.32 $\pm$ 3.87 | 110.39 $\pm$ 0.93 | 110.13 $\pm$ 0.21 | 109.07 $\pm$ 1.30 | 99.62 $\pm$ 13.55 | 58.77 $\pm$ 15.75 | 75.16 $\pm$ 23.40 | 108.51 $\pm$ 0.60 |
> > > > | lift-ph             | 84.75 $\pm$ 6.23  | 91.50 $\pm$ 5.42  | 91.75 $\pm$ 5.90  | 98.84 $\pm$ 2.33  | 96.6 $\pm$ 2.96   | 85.2 $\pm$ 7.71   | 98.40 $\pm$ 2.59  | 97.60 $\pm$ 2.94  |
> > > > | lift-mh             | 91.00 $\pm$ 2.82  | 90.75 $\pm$ 5.75  | 86.75 $\pm$ 5.95  | 90.04 $\pm$ 4.45  | 60.40 $\pm$ 25.11 | 52.60 $\pm$ 10.07 | 89.00 $\pm$ 4.37  | 87.20 $\pm$ 5.31  |
> > > > | can-ph              | 68.00 $\pm$ 9.13  | 62.00 $\pm$ 10.90 | 69.67 $\pm$ 5.89  | 76.40 $\pm$ 3.67  | 63.2 $\pm$ 11.86  | 25.4 $\pm$ 5.25   | 68.6 $\pm$ 7.66   | 74.8 $\pm$ 2.40   |
> > > > | can-mh              | 47.50 $\pm$ 3.51  | 30.50 $\pm$ 8.73  | 50.50 $\pm$ 6.48  | 53.6 $\pm$ 7.86   | 30.4 $\pm$ 23.03  | 13.8 $\pm$ 5.73   | 59.0 $\pm$ 5.0    | 57.6 $\pm$ 5.00   |
> > > >
> > > >
> > > >  We hope these additional experiments can help address the reviewers other concerns about empirical evaluation.
> > > >
> > > > Thank you and please let us know if there are other concerns.

---

### Official Review · Reviewer_HQUc · 2023-07-05

**Soundness:** 3 good
**Presentation:** 3 good
**Contribution:** 3 good
**Rating:** 7
**Confidence:** 4

**Summary:**

This paper addresses the problem of learning a policy given offline pairwise preference data. The proposed algorithm, inverse preference learning, directly learns a value function without explicitly learning the reward function first. This setup simplifies the common two-step approach of first inferring the reward function in a supervised manner and then learning the policy using RL. Across a range of simulated robotics experiments, it is shown that this method performs at least as well as more complex approaches that have significantly more (hyper-)parameters.

**Strengths:**

The paper is well-motivated and clearly written. The presented idea is novel, simple and well-executed. The resulting algorithm is easy to implement, has much fewer (hyper-)parameters compared to prior work, and seems to perform on par or better than prior work. Source code is also available. Given the increasing interest in reinforcement learning with human feedback (RLHF) for robotics and natural language problems, this paper should be of high interest for the community.

Finally, I really appreciate that the authors invested time to improve the Markovian Reward (MR) baseline. This by itself is a nice little contribution, and I hope that future work will adopt this much stronger baseline.

**Weaknesses:**

The presentation of the results in the tables could be improved. While in Table 1 the results are based on the final success rate at the end of training, it seems that the results in Table 2 are based on the best success rate achieved throughout training (particularly apparent for the Drawer open tasks). I found this to be a bit misleading, especially since IPL seems to heavily suffer from the problem that the success rate deteriorates over time. Furthermore, I didn't fully understand why some results in the tables are bold and others are not. For example, in the Assembly task with 500 queries (Table 2), it seems that IPL is not significantly better than MR.

Minor comments:
1. I initially got a bit confused as to why $L_r$ depends on $D_o$ and $D_p$. Perhaps you could remind the reader that the preferences in $D_p$ are also (sub-)trajectories.
2. Typos:
- line 89: "offlien" -> "offline"
- line 101: "Our work build on" -> "Our work builds on"
- Eq. 3: Subscripts t missing for $a, s$ in $\pi$ and $\mu$
- Eq. 4: $p(\cdot|s|a)$ -> $p(\cdot|s,a)$
- line 158: "to learned" -> "to the learned"
- line 199: "KL-constrianed" -> "KL-constrained"
- line 223: "thus" -> "is [thus]"
- line 283: "that use" -> "that uses"
- line 305: "Data" -> "data"
- line 315: "form" -> "from"
- line 308/319: "Preference-based" -> "preference-based"

**Questions:**

1. Could you please clarify the results in Table 2? I would suggest that both Table 1 and Table 2 either only report the final success rate or report both the final and best success rate.
2. Could you please double-check/clarify why certain numbers in the results are bold and others are not?

**Limitations:**

One of the original motivations of inverse RL was that the reward function provides a succinct and transferable definition of the task (Abbeel and Ng, 2004). While recent approaches seem to no longer learn "succinct" reward functions, one could argue that transferability is still a good reason to explicitly learn a reward function. Thus, a potential limitation of the presented approach is that the ability to use the learned reward function to learn different policies gets lost since the value function depends on a particular policy.

---

> ### Author Rebuttal · Authors · 2023-08-10
>
> We would like to thank the reviewer for their in-depth review. We hope to have addressed all the reviewers' concerns, and would invite them to additionally examine the new experimental and theoretical results produced in response to other reviewers.
>
> **Clarification of Results in Table 2**
>
> In our uploaded response page we have included evaluations at a fixed stopping point, like Table 1, and find the trend to be similar: IPL and MR+IQL perform very similarly, with IPL performing slightly better in 3 of the 5 tasks. We will change this to the main version of the Table in the final version. We hope this improves the consistency of our experiments.
>
> **Bolding Scheme**
>
> Following IQL [1], we bold values when they are within 95% of the best method. We will state this explicitly in all Table captions. For the specific example brought up by the Reviewer, in Table 2 on Assembly for 500 queries IPL gets 0.9, 95% of which is 0.855, so MR would not be bolded. We understand that this scheme has less significance when performance values are quite low, and can make note of as well in the Table caption.
>
> **Limitations**
>
> Thank you for pointing this out! We agree that there are some settings where transferring a reward function may be more useful than transferring a Q-function – largely stemming from the fact that a Q-function is tied to an inherent policy. If one wants to collect sub-optimal data for a task, a reward function may be necesary. Many Meta-RL approaches are also designed to leverage reward functions instead of Q-functions. Reward functions learned from preferences could also be used to smooth sparse reward functions – a Q-function could not easily be used for smoothing (though the advantage could! Ng. 1999). We will note the additional limitation of the transferability of an implicit reward function in Section 5 and include this discussion.
>
> Thank you for also finding typos! We have fixed them.

---

> > ### Comment · Reviewer_HQUc · 2023-08-18
> >
> > Thank you for your response. I appreciate the improved presentation in the paper as well as improved clarification through the rebuttal. On the other hand, I wasn't aware that some of the ideas used in the paper have already been introduced in prior work (as brought up by other reviewers). Overall, both aspects cancel each other out, and thus I will maintain my original score.

---

> > > ### Author Response · Authors · 2023-08-18
> > > **Thank you for responding.**
> > >
> > > Dear Reviewer,
> > >
> > > We appreciate your engagement in the discussion period! We are happy to have addressed your concerns by improving the presentation of the paper and clarification through the rebuttal. We will make sure to propagate your comments to the final draft.
> > >
> > > Though the soft-inverse bellman operator was introduced in prior work (IQ-Learn), it has only been used in the context of online imitation learning, and has not yet been used for learning from preferences. We believe that showing the application of the (general) inverse bellman operator to the preference-based RL framework is a strong and non-trivial contribution. First, preference-based RL has become increasingly popular because of its ability to align policies / models with human intent, which will be important for robots deployed in the real world. Second, IPL shows that we can greatly simplify the PbRL problem under any RL algorithm and make learning more efficient without degrading performance. Doing this required both theoretical and engineering innovations.
> > >
> > > Thank you for your consideration!

---

### Official Review · Reviewer_kKfF · 2023-07-05

**Soundness:** 2 fair
**Presentation:** 3 good
**Contribution:** 3 good
**Rating:** 6
**Confidence:** 3

**Summary:**

This paper proposes a new algorithm, inverse preference learning, to learn from (offline) preferences between behavior segments. In particular, they show that human preferences can be modeled using only the Q-function, therefore eliminating the need to learn a separate reward function. The authors show this approach can succeed in a variety of environments.

**Strengths:**

This paper has several strengths:
- They study the important problem of preference-based RL.
- This paper makes the novel insight that human preferences can be modeled using only the Q-function, and introduce an algorithm based on this insight.
- They experimentally validate different aspects of their algorithm over a wide variety of tasks.

**Weaknesses:**

**W1.** The motivation that reward-modeling has a large cost is a little bit weak (especially with regards to memory). For example, InstructGPT uses a 6B reward model and a 175B policy. The reward model is therefore only a small fraction of the memory cost. In addition, one of the main hyperparameters of reward-modeling that the authors say is difficult to tune is the stopping criteria. However, the stopping criteria for reward model training is straightforward in most papers: Simply monitor the validation loss on a held-out fraction of comparison data.

**W2.** This paper mainly focuses on offline RL, yet the most impactful RLHF applications are trained online, where behavior data is collected from the policy. It seems like IPL can be trained in an online manner as well, but it is not clear from the presentation.

**Questions:**

**Q1.** Can this method be used in an online setting, where new data is generated according to the policy?

**Q2.** How do MR and IPL perform without data augmentation? And why does MR receive such a large boost based on this augmentation?

**Q3**. Are the reimplemented training setup and the training setup from [25] exactly the same? If not,  PT and LSTM should be reimplemented for fair comparison.

**Q4.** What is the bolding scheme in Table 1?

**Q5.** Is there any way to experimentally validate the claim that IPL is more hyperparameter efficient? For example, random/grid hyperparamter search with an equivalent budget? From Figure 2, it looks like MR is actually more hyperparameter efficient than IPL.

**Limitations:**

Yes, the authors do discuss limitations of their work.

---

> ### Author Rebuttal · Authors · 2023-08-10
>
> We would like to thank the reviewer for their comments.
>
> **Weakness 1: Reward modeling cost**
>
> While reward models in RLHF in language models can be smaller, as in InstructGPT in control-based domains, the current trend is the opposite: reward models are getting bigger. Kim et al. [1] and Early et al [2] use reward models that are significantly larger than the policy and/or critic. In control, we often don’t have access to large amounts of applicable pre-training data, and thus the quality of the reward function is paramount to attaining good success. Moreover, while reward models in RLHF, like those in InstructGPT, provide feedback like a single-step contextual bandit at the end of generation, reward functions for control need to provide feedback at every step, analogous to “at each token”.
>
> Using validation loss as a stopping criterion for reward models as the reviewer suggests, while potentially accurate, can be expensive in practice. Human preferences in many domains (including control/robotics domains) are quite expensive to collect. In addition these preferences are often high-variance, requiring an even larger validation dataset to account for the high variance.
> In addition, we note that IPLremoves the inherent hyper-parameters in reward modeling, such as the network architecture.
>
> **Weakness 2: Online RL**:
>
> We agree that the online setting is also interesting, and there is nothing stopping IPL from being used online since it can be combined with any off-policy RL algorithm. To test an online version of IPL, we combine our framework with SAC and compare it to PEBBLE, a standard, state-of-the-art method for off-policy online Preference-based RL [3]. We include learning curves in Figure 1 of the attached document. On two of the harder Meta-World tasks from Table 2, plate-slide and drawer-open, we find that IPL trades blows with PEBBLE (out performing by a large margin on drawer-open) while exhibiting lower variance.
>
> **Questions**
>
> *Q1: Online Settings*. Yes! See our response to Weakness 2.
>
> *Q2: Data Augmentation*:  We ablate the use of augmentations for both IPL and MR+IQL in Hopper and Robomimic Can in Figure 2 of our uploaded page. We find that data-augmentation makes a substantial difference for MR+IQL on Hopper, but has a smaller effect in the robotics domains. The reviewer asks why this form of data-augmentation makes such a big difference. We would like to point out that this question was previously studied in SURF [5]. The fact that humans likely made similar judgements between subsets of segments is a good inductive bias for the reward function, helping its accuracy when relabeling the offline dataset.
>
>
> *Q3: Implementation*: We use the same exact datasets as [1] and the same evaluation procedure. We use the same exact network architectures, learning rates (and schedules). The primary difference in our implementation is the addition of data-augmentation, which we believe should be as it has already been shown to be effective in SURF [5]. Due to the use of data-augmentation, we also adjusted the reward training steps. Everything else remains identical. The data-augmentation techniques are inapplicable to PT [1] and the NMR baseline from [2], since they are sequence modeling approaches which require the whole data sequence.
>
> We would like to additionally point out that Reviewer HQUc views our improved MR baseline to be “itself a nice little contribution”.
>
> *Q4: Bolding Scheme*: We bold results within 95% of the best performing method, as done in IQL [4]. We will include this clarification in all Table Captions.
>
> *Q5: Hyper-parameter Efficiency*: The reviewer asked us to further discuss the hyper-parameter efficiency of IPL. The only parameter we searched over when tuning IPL was the regularization coefficient. We include additional ablations on the regularization coefficient $\lambda$ in our attached page, showing that in many cases, IPL is very robust to this value. We would also like to again highlight that  reward-modeling methods also have to choose an architecture for the reward network, its learning rate, batch size, etc. in addition to the stopping point parameter we ablate for MR+IQL and thus Figure 2 is not the complete picture. Nonetheless, we will reward our points surrounding hyper-parameter efficiency.
>
> [1] Kim et al.. Preference transformer. ICLR 2023.
>
> [2] Early et al.. Non-markovian reward modeling from trajectory labels. NeurIPS 2022.
>
> [3] Lee et al., PEBBLE. ICLR 2021.
>
> [4] Ilya Kostrikov et al.. Offline RL with implicit q-learning.ICLR, 2022.
>
> [5] Park et al. SURF: Semi-supervised reward learning with data augmentation. ICLR 2022.

---

> > ### Comment · Reviewer_kKfF · 2023-08-18
> > **Response by Reviewer**
> >
> > Thank you for the detailed response! My response have largely been addressed. I will adjust my score accordingly.

---

### Official Review · Reviewer_ggMH · 2023-07-07

**Soundness:** 3 good
**Presentation:** 3 good
**Contribution:** 3 good
**Rating:** 7
**Confidence:** 4

**Summary:**

RLHF pipelines typically consist of (1) training a reward model over human preference data and (2) using this trained reward model with a well-known RL method. This two stage training is computationally expensive. The authors of this paper develop an algorithm "Inverse Preference Learning" to directly learn the $Q, V$ functions, which can be easily used to extract an aligned policy. By directly learning the value functions, this approach bypasses learning a reward model, which is expensive to train, and prone to problems like reward hacking.

**Strengths:**



**Weaknesses:**



**Questions:**

On Page 2, the authors mention - "This can be problematic as prediction errors cascade from the reward function, to the critic, and ultimately the actor causing high variance in downstream performance." Can we quantify these prediction errors more formally, to establish how serious this issue is?

Only having two options in the preference data seems limiting. Can this idea be applied to the case where the preference is amongst more than two options? Eg. maximizing $p(a\succ b\succ c\succ d)$ rather than $p(a\succ b)$.

On Page 6, line 214, the authors say "While such a reward function seems unrealistic", here, why do we want rewards to be necessarily continuous? In many cases, human defined rewards are discontinuous (eg. 0-1 type of reward where the reward is 0 everywhere except when the agent reaches the goal, when it is 1).

The authors mention on Line 53-54 that "the key insight of our work is that, under a fixed policy, the Q-function learned by off-policy RL algorithms captures the same information as the learned reward function". This is not an insight developed in this paper, but rather a result from [1] (see Lemma 3.2 in [1]). Moreover, the result is for soft-Q functions, and not the standard Q function. I would recommend to make these points clear in the manuscript.

Finally, there is parallel work [2] that also tries to bypass reward learning similar to this approach. It might be good to acknowledge, but I leave this decision to the authors.

Suggestions related to language and typos:
1. Page 2, 2nd last line - "offlien" -> "offline"
2. Page 6, 3rd line - should "another perturbation of size \epsilon" be "another perturbation of size \epsilon prime"?

References:
1. IQ-Learn: Inverse soft-Q Learning for Imitation, Garg et al. (2022)
2. Direct Preference Optimization: Your Language Model is Secretly a Reward Model, Rafailov et al. (2023)

EDIT (16 Aug 2023): Updated score from 6->7.

**Limitations:**

---

> ### Author Rebuttal · Authors · 2023-08-10
>
> Thank you for your detailed review! The reviewer largely had theoretical questions about our work. We believe the answers to these questions will help all readers, and will correspondingly update the paper to include all the information below.
>
> **1. Cascading Errors**
>
> The reviewer asked if we could better characterize the cascading error problem. First, cascading errors in Preference-based RL can be empirically observed in many prior works. For example, in PEBBLE [1] Figure 4, we see that confidence bands for preference-based methods are far larger than those of SAC with oracle rewards. Second, we can actually theoretically characterize this problem using arguments from [2] designed for behavior cloning. In their work, once an error is made a policy goes out of distribution and will subsequently only make errors. This “cascading” error framing can be applied to the networks used in PbRL. Assume that a network makes a prediction error with probability $\epsilon$, and that all future networks necessarily make a prediction error when a previous one does, i.e. if the reward model makes a mistake, then the critic and actor also make a mistake. By [2] the total error across all networks can be bounded as $O(\epsilon N^2)$, where $N$ is the number of networks. By reducing the number of networks by 1, IPL can *theoretically* lower this bound.
>
> **2. Only Two Forms of Data**
>
> The reviewer stated that having two options in the preference data is limiting. We presented IPL only for binary preferences as it is the simplest case, but IPL can easily be extended to rankings using a Plackett Luce Model. Consider permutations $\tau$ over $K$ segments:
>
> $P_{r_E}(\tau) = \prod_{k=1}^K  \left(\exp \sum_t r_E(s^{\tau_k}_t, a^{\tau_k}_t)\right) / d_k$
>
> where $d_k = \sum_{j=k}^K \exp \sum_{t} r_E(s^{\tau_j}_t, a^{\tau_j}_t)$
>
> Then, we make the same substitution using the inverse bellman operator giving us the permutation model implied by the Q function, and run maximum likelihood estimation over the model.
>
> $L_p (r) = \mathbb{E}_{\tau \sim \mathcal{D}_p }\left[ \log P_r (\tau) \right]$
>
> We will include a full derivation of IPL for rankings in the Appendix.
>
> **3. Regularizing the Reward Function**
>
> The reviewer asked why we might want reward functions to be smooth instead of discontinuous, as in binary. We hypothesize that, though a human may design a binary reward function, human preferences are often smooth. For example, we might make judgments more smoothly across different comparisons, instead of attributing all reward to a single frame. Moreover, smooth reward functions have been shown to generally perform better [3], while sparse ones are harder to optimize.
>
>  In our case the regularization is also necessary to remove the ambiguity of the Bradley Terry model – which can only recover $r_E$ up to a constant. See proof in the central reviewer response.
>
> **4. Key Insight**
>
> Thank you for pointing this out – we intended to frame our key insight as extending the inverse-bellman operator from IQ-Learn. We will update the text to make it clear that our insight is applying this to the reward function in preference-based RL.
>
> **Soft-Q vs. Standard, Proof of Bijection**
>
> While we presented our method (IPL) with XQL, IPL can in fact use any RL update. We thought that presenting our method with XQL would make it easier to understand in the context of IQ-Learn, which originally introduced the inverse bellman operator and used soft Q-learning. IQ-Learn requires the soft-Q framework because a) it was developed for soft-inverse RL, and b) it guarantees that the saddle-point between the reward and policy is unique for inverse-RL. We used IQL in experiments to match baselines, and because XQL is often unstable.
>
> However, the bijection between Q-functions and reward functions under a fixed policy exists regardless if one uses soft-Q learning, allowing IPL to use any RL update. We will make this more clear in the final draft. Lemma 3.2 in IQ-Learn in fact does not depend on the soft-Q framework:
>
> Let $P^\pi$ be the transition matrix for the MDP corresponding to a fixed policy $\pi$. In vector form, the bellman equation becomes $Q = r + \gamma P^\pi Q$, or $r = (I - \gamma P^\pi) Q$. We can establish a bijection by showing that $(I - \gamma P^\pi)$ is invertible. $||\gamma P^\pi|| < 1$ by construction, as $P^\pi$ is bounded by 1 as a probability distribution and $\gamma < 1$, which guarantees that its Neumann series converges. This implies the existence of $(I - \gamma P^\pi)^{-1}$. Thus, $Q = (I - \gamma P^\pi)^{-1} r$ and a bijection exists.
>
> **Connections to DPO**
>
> We are aware of the recent DPO work, which was released after the NeurIPS submission deadline. While IPL and DPO share some theoretical connections, we’d like to note that DPO is designed for and limited to the contextual bandit setting. This setting is appropriate for RLHF in LLMs; however, DPO would not apply to the more general setting of preferences over sequences of states and actions. IPL is strictly more general, as if you take IPL with XQL for bandits, you can exactly recover DPO.
>
> Within the bandits setting, there is no  “next-state” and $V^*(s’)$ is removed, and the inverse bellman operator becomes just $Q(s,a) = r(s,a)$. The optimal XQL policy is $\pi^* = \mu(a|s) e^{Q^*(s,a)}/ Z(s)$ where $Z$ is the partition function. By rearranging, $TQ = Q^*(s,a) = \log \frac{\pi(a|s)}{\mu(a|s)} + Z(s)$. We can plug this into the preference model induced by Q in Eq 6 of IPL. In the RLHF setting, the partition function cancels since we assume the context to be the same between preferences. This exactly results in the DPO algorithm, showing the DPO is in fact just an instantiation of IPL. We will include this in the Appendix.
>
> Thanks for finding typos!
>
> [1] Lee et al., PEBBLE. ICLR 2021.
>
> [2] Ross and Bagnell. Efficient reductions for imitation learning, 2010.
>
> [3] Ng, A. et al. Policy invariance under reward transformations ICML, 1999.

---

> > ### Comment · Reviewer_ggMH · 2023-08-16
> > **Rebuttal response**
> >
> > The author's response has answered most of my queries. I will increase my score to 7.

---

### Author Rebuttal · Authors · 2023-08-10

Dear reviewers,

Thank you all for your detailed feedback. We have responded to all reviewers individually with more content but wanted to make a global list of the major changes that we have made to the manuscript and additional experiments included in our allowed one page upload.

**Theoretical**

A number of reviewers had questions about the theory and derivation of IPL. We have made a number of changes to improve our understanding of the method.

1. Reviewers ggMH and HK3n asked about why IPL was derived with XQL but our experiments were run with IQL. Originally, we used XQL for our derivations to be consistent with the Inverse RL literature that has previously used the MaxEnt Framework, but used IQL for experiments to exactly match baselines. To remove this confusion, we have updated the method section with a new expanded derivation of IPL under any off-policy RL algorithm that works via policy evaluation and policy improvement steps, not just XQL.

2. We apologize if the contribution statement was not clear (Reviewers ggMH, HK3n). We have now made it clear that our contribution is the application of the inverse bellman operator to the Preference-based RL, not the inverse-bellman operator itself.

3. Reviewers ggMH and HK3n asked about the validity of IPL’s practical implementation given it does not necessarily use a MaxEnt RL algorithm. However, the bijection between $Q$ and $r$ for a fixed policy holds in the general case (not just soft Q-Learning). In our response to Reviewer ggMH we include a proof of this bijection, analogous to Lemma 3.2 in IQ-Learning, for the general case.

4. Reviewer HK3n asked for a more theoretical understanding of IPL, which we believe will be of interest to all reviewers. We prove that IPL converges to the optimal policy for the expert reward function subject to regularization and show that regularization is necessary to guarantee this convergence. For space, this is included below and will be in the final version of the paper.




**Experimental**

See uploaded pdf

1. Reviewers kKfF asked about IPL’s applicability to online settings. Since IPL works for any Off-Policy RL algorithm, it also works online. We compare it with PEBBLE on two of the harder Meta-World tasks from Table 2, and find that IPL trades blows with PEBBLE while exhibiting lower variance.

2. Reviewers kKfF asked about our implementation of MR+IQL and data-augmentation. We ablate the use of augmentations for both IPL and MR+IQL in Hopper and Robomimic Can. We find that data-augmentation makes a substantial difference for MR+IQL on Hopper, but has a smaller effect in the robotics domains.

3. Reviewers kKfF and HK3n asked about the regularization values used by IPL. We include additional ablations on the regularization coefficient $\lambda$, showing that in many cases, IPL is very robust to this value.

4. Reviewer HQUc asked that we change Table 2 to use the same evaluation criterion as Table 1. We have included evaluations at a fixed stopping point and find the trend to be similar: IPL and MR+IQL perform very similarly, with IPL performing slightly better in 3 of the 5 tasks.

**Proof that IPL Converges to the Optimal Policy corresponding to the regularized expert reward**

First, note that *the Bradley-Terry Model is underspecified* and only recovers $r_E$ up to a constant shift, as constants are canceled by the Boltzmann. While constant shifts don’t change the optimal policy, it does change the Q-function. To prove convergence, we show that the sequence of Q-functions is increasing, as is standard practice for off-policy RL algorithms. This is only possible if the reward function does not shift when optimizing the preference loss.

We prove this statement in the Tabular setting. Let $Q_t \in \mathbb{R}^{|S \times A|}$ and $\pi_t$ indicate the Q-function and policy after update $t$, respectively. Let $Q_0 = 1/(1 - \gamma) \min_{S \times A} r(s,a)$. The inverse bellman operator tells us, in vector form, that $r = (I - \gamma P^\pi)Q$  where $P^\pi$ is the transition matrix. Let $r^* = \arg \min_r \mathbb{E}_{D_p}[y \log P_r + (1-y) \log (1 - P_r) + \lambda r^2]$, or the minimizer of the regularized preference loss.

At each step of IPL, we substitute the inverse bellman operator into the preference loss and optimize. Thus at convergence, $ (I - \gamma P^{\pi_t})Q_t = r^*$ uniquely because of the bijection between $r$ and $Q$ under a fixed $\pi$ (See proof under rebuttal to Reviewer ggMH Soft Q vs Standard).

Then, we use any off-policy RL algorithm that guarantees convergence (ie $Q^{\pi_{t+1}} \geq Q^{\pi_{t}}$) to obtain a new policy $\pi_{t+1}$ from $\pi_t$ and $Q_t$. Using $\pi_{t+1}$ we can obtain the transition matrix $P^{\pi_{t+1}}$ in tabular settings.

Finally, we optimize the preference loss again using $P^{\pi_{t+1}}$ in the inverse Bellman operator to obtain $Q_{t+1}$. At convergence $ (I - \gamma P^{\pi_{t+1}})Q_{t+1} = r^*$ holds.

Due to regularization $r^*$ is unique, and thus $Q_t$ and $Q_{t+1}$ are both Q-functions for the reward function $r^*$, just under different policies. If we did not have regularization, this might not be the case. However, we know from the policy improvement step, that $Q^{\pi_{t+1}} \geq Q^{\pi_{t}}$ necessarily, and thus $Q_{t+1} \geq Q_{t}$ for any $t$. The base case holds because $ \forall \pi, Q_0 \leq Q^\pi$ by construction. This concludes the proof.

Thus, by repeatedly optimizing the Preference Loss and improving the policy, we obtain the optimal policy corresponding to $r^*$.

This can also be seen (very) informally by looking at Eq. 6 in our paper: If the policy improvement step works, then $V^{\pi_{t+1}}(s’) \geq V^{\pi_{t}}(s’)$, and thus the $Q(s,a)$ will need to *increase* to fit the same optima of the regularized preference loss.

**Conclusion**

We hope that the increased theoretical and experimental rigor addresses the reviewers concerns. See individual responses for more clarifications.

Thank you!

---

> ### Author Response · Authors · 2023-08-14
> **Addendum to proof:**
>
> Thank you to reviewer HK3n for point out that our original language was not clear. This proof requires $r^*$ to be unique. Thus, we assume access to sufficient preference data and any general regularizer $\psi$ such that $r^* = \arg \min_r \mathbb{E}_{D_p}[y \log P_r + (1-y) \log (1 - P_r) + \lambda \psi(r)]$ is unique. Without access to sufficient preference data, the standard, un-regularized BCE preference loss objective does not recover $r_E$. Even with sufficient data, the standard preference BCE loss objective recovers $r_E$ up to a constant shift.

---

> ### Author Response · Authors · 2023-08-16
> **Additional Experiments**
>
> In response to Reviewer HK3n, we have added additional baselines and copied the results here for convenience.
>
> 1. BREX (Safe Imitation Learning via Fast Bayesian Reward Inference from Preferences).
> 2. IPL + XQL. Our method with the XQL learning algorithm as originally presented in the method section. Note that performance crashed due to instability on walker2d-expert.
> 3. IPL without regularization. This shows how regularization is important to empirical performance.
> 4. We would also like to note that TREX with IQL instead of PPO applied to our datasets is the same as MR + IQL.
>
> |                     | MR                | LSTM              | PT                | MR (reimpl)       | BREX              | IPL Without Reg.  | IPL+XQL           | IPL+IQL           |
> |---------------------|-------------------|-------------------|-------------------|-------------------|-------------------|-------------------|-------------------|-------------------|
> | hopper-med-replay   | 11.56 $\pm$ 30.27 | 57.88 $\pm$ 40.63 | 84.54 $\pm$ 4.07  | 70.20 $\pm$ 35.0  | 62.0 $\pm$ 20.29  | 10.41 $\pm$ 2.26  | 80.4 $\pm$ 2.13   | 73.57 $\pm$ 6.67  |
> | hopper-med-expert   | 57.75 $\pm$ 23.70 | 38.63 $\pm$ 35.58 | 68.96 $\pm$ 33.86 | 102.97 $\pm$ 5.55 | 85.1 $\pm$ 7.97   | 52.81 $\pm$ 7.45  | 54.3 $\pm$ 12.33  | 74.52 $\pm$ 10.11 |
> | walker2d-med-replay | 72.07 $\pm$ 1.96  | 77.00 $\pm$ 3.03  | 71.27 $\pm$ 10.30 | 68.79 $\pm$ 5.64  | 10.30 $\pm$ 5.44  | 4.85 $\pm$ 1.52   | 57.82 $\pm$ 5.24  | 59.92 $\pm$ 5.11  |
> | walker2d-med-expert | 108.32 $\pm$ 3.87 | 110.39 $\pm$ 0.93 | 110.13 $\pm$ 0.21 | 109.07 $\pm$ 1.30 | 99.62 $\pm$ 13.55 | 58.77 $\pm$ 15.75 | 75.16 $\pm$ 23.40 | 108.51 $\pm$ 0.60 |
> | lift-ph             | 84.75 $\pm$ 6.23  | 91.50 $\pm$ 5.42  | 91.75 $\pm$ 5.90  | 98.84 $\pm$ 2.33  | 96.6 $\pm$ 2.96   | 85.2 $\pm$ 7.71   | 98.40 $\pm$ 2.59  | 97.60 $\pm$ 2.94  |
> | lift-mh             | 91.00 $\pm$ 2.82  | 90.75 $\pm$ 5.75  | 86.75 $\pm$ 5.95  | 90.04 $\pm$ 4.45  | 60.40 $\pm$ 25.11 | 52.60 $\pm$ 10.07 | 89.00 $\pm$ 4.37  | 87.20 $\pm$ 5.31  |
> | can-ph              | 68.00 $\pm$ 9.13  | 62.00 $\pm$ 10.90 | 69.67 $\pm$ 5.89  | 76.40 $\pm$ 3.67  | 63.2 $\pm$ 11.86  | 25.4 $\pm$ 5.25   | 68.6 $\pm$ 7.66   | 74.8 $\pm$ 2.40   |
> | can-mh              | 47.50 $\pm$ 3.51  | 30.50 $\pm$ 8.73  | 50.50 $\pm$ 6.48  | 53.6 $\pm$ 7.86   | 30.4 $\pm$ 23.03  | 13.8 $\pm$ 5.73   | 59.0 $\pm$ 5.0    | 57.6 $\pm$ 5.00   |

---

### Decision · Program_Chairs · 2023-09-21

**Decision:**

Accept (poster)

**Comment:**

The reviews are certainly in favor. The results are promising. However, according to the reviews, a few aspects needs further improvements. First of all, both of the proposed method and its proof could use a more thorough and delicate discussion of related work. A few reviews mention its similarity to IQLearn and other prior work, and hence challenge this paper's novelty. The rebuttal does not convince most reviews into changing their opinion on this. Why the changes are needed, and what are the benefits could be better explained. We would like this paper to incorporate most of the related work mentioned in reviews in its final version. Secondly, a more thorough discussion of its limitation is needed. For example, as mentioned in a few reviews. There is definitely cost associated with no explicit reward function. What is the cost (e.g., transferability, learning stability), and why this trade-off is appealing in certain applications? Overall, most reviews believe this is a solid paper, and its results outweigh the concern over its novelty.